# FACTORING OUT PRIOR KNOWLEDGE FROM LOW-DIMENSIONAL EMBEDDINGS

## ABSTRACT

Low-dimensional embedding techniques such as tSNE and UMAP allow visualizing high-dimensional data and therewith facilitate the discovery of interesting structure. Although they are widely used, they visualize data as is, rather than in light of the background knowledge we have about the data. What we already know, however, strongly determines what is novel and hence interesting. In this paper we propose two methods for factoring out prior knowledge in the form of distance matrices from low-dimensional embeddings. To factor out prior knowledge from tSNE embeddings, we propose JEDI that adapts the tSNE objective in a principled way using Jensen-Shannon divergence. To factor out prior knowledge from any downstream embedding approach, we propose CONFETTI, in which we directly operate on the input distance matrices. Extensive experiments on both synthetic and real world data show that both methods work well, providing embeddings that exhibit meaningful structure that would otherwise remain hidden.

## 1 INTRODUCTION

Embedding high dimensional data into low dimensional spaces, such as with tSNE (van der Maaten & Hinton, 2008) or UMAP (McInnes et al., 2018), allow us to visually inspect and discover meaningful structure from the data that would otherwise be difficult or impossible to see. These methods are as popular as they are useful, but, at the same time limited in that they are one-shot only: they embed the data as is, and that is that. If the resulting embedding reveals novel knowledge, all is well, but, what if the structure that dominates it is something we already know, something we are no longer interested in, or, if we want to discover whether the data has meaningful structure other than what the first result revealed? In word embeddings, for example, we may already know that certain words are synonyms, while in single cell sequencing we may want to discover structure other than known cell types, or factor out family relationships. The question at hand is therefore, how can we obtain low-dimensional embeddings that reveal structure *beyond* what we already know, i.e. how to factor out prior knowledge from low-dimensional embeddings?

For conditional embeddings, research so far mostly focused on *emphasizing* rather than factoring out prior knowledge (De Ridder et al., 2003; Hanhijärvi et al., 2009; Barshan et al., 2011), with conditional tSNE as notable exception, which, however, can only factor out label information (Kang et al., 2019). Here, we propose two techniques for factoring out a more general form of prior knowledge from low-dimensional embeddings of arbitrary data types. In particular, we consider background knowledge in the form of pairwise distances between samples. This formulation allows us to cover a plethora of practical instances including labels, clustering structure, family trees, user-defined distances, but also, and especially important for unstructured data, kernel matrices.

To factor out prior knowledge from tSNE embeddings, we propose JEDI, in which we adapt the tSNE objective in a principled way using Jensen-Shannon divergence. It has an intuitively appealing information theoretic interpretation, and maintains all the strengths and weaknesses of tSNE. One of these is runtime, which is why UMAP is particularly popular in bioinformatics. To factor out prior knowledge from embedding approaches in general, including UMAP, we hence propose CONFETTI, which directly operates on the input data. An extensive set of experiments shows that both methods work well in practice and provide embeddings that reveal meaningful structure beyond provided background knowledge, such as organizing flower images according to shape rather than color, or organizing single cell gene expression data beyond cell type, revealing batch effects and tissue type.

## 2 RELATED WORK

Embedding high dimensional data into a low dimensional spaces is a research topic of perennial interest that includes classic methods such as principal component analysis Pearson (1901), multi-dimensional scaling (Torgerson, 1952), self organizing maps (Kohonen, 1982), and isomap (Tenenbaum et al., 2000), all of which focus on keeping large distances intact. This is inadequate for data that lies on a manifold that resembles a Euclidean space only locally, which is the case for high dimensional data (Silva & Tenenbaum, 2003) and for which we hence need methods such as locally linear embedding (LLE) (Roweis & Saul, 2000) and stochastic neighbor embedding (SNE) (Hinton & Roweis, 2003) that focus on keeping local distances intact. The current state of the art methods are t-distributed SNE (tSNE) by van der Maaten & Hinton (2008) and Uniform Manifold Approximation (UMAP) by McInnes et al. (2018). Both are by now staple methods for data processing, e.g. in biology (Becht et al., 2019; Kobak & Berens, 2019) and NLP (Coenen et al., 2019). As they often yield highly similar embeddings (Kobak & Linderman, 2019) it is a matter of taste which one to use. While tSNE has an intuitive interpretation, despite recent optimizations (van der Maaten, 2014; Linderman et al., 2019) compared to UMAP it suffers from very long runtimes.

Whereas the above consider only the data as is, there also exist proposals that additionally take user input and/or domain knowledge into account. For specific applications to gene expression, it was proposed to optimize projections of gene expression to model similarities in corresponding gene ontology annotations (Peltonen et al., 2010). More recently, attention has been brought to removing unwanted variation (RUV) from data using negative controls in particular in the light of gene expression, assuming that the expression can be modeled as a linear function of factors of variation and a normally distributed variable (Gagnon-Bartsch & Speed, 2012). This approach has been successfully applied to different tasks and domains of gene expression (Risso et al., 2014; Buettner et al., 2015; Gerstner et al., 2016; Hung, 2019). Here, we are interested to develop a domain independent method to obtain low-dimensional embeddings while factoring out prior knowledge. For that, we neither want to assume a functional relationship between prior and input data, nor do we want to assume a particular distribution of the input, but keep the original data manifold intact. Furthermore, we do not want to rely on negative samples that have to be known and present in the data to be able to factor out the prior.

The general, domain independent methods supervised LLE (De Ridder et al., 2003), guided LLE (Alipanahi & Ghodsi, 2011), and supervised PCA (Barshan et al., 2011) all aim to emphasize rather than factor out the structure given as prior knowledge. Like us, Kang et al. (2016; 2019); Puolamäki et al. (2018) factor out background knowledge, but are much more limited in the type of prior knowledge. In particular, Puolamäki et al. (2018) requires users to specify clusters in the embedded space, Kang et al. (2016) requires background knowledge for which a maximum entropy distribution can be obtained, while Kang et al. (2019) extend tSNE and propose conditional tSNE (ctSNE) which accepts prior knowledge in the form of class labels. In contract, we consider prior knowledge in the form of arbitrary distance metrics, which can capture relative relationships which appears naturally in real world data, such difference in age, geographic location, or level of gene expression. We propose both, an information theoretic extension to tSNE, and an embedding-algorithm independent approach to factor out prior knowledge.

## 3 THEORY

We present two approaches, with distinct properties, that both solve the problem of embedding high dimensional data while factoring out prior knowledge. We start with an informal definition of the problem, after which we introduce vanilla tSNE. We then present our first solution, JEDI, which extends the tSNE objective to incorporate prior information. We then present CONFETTI, which uses an elegant yet powerful idea that allows us to directly factor out prior knowledge from the distance matrix of the high dimensional data, which allows CONFETTI to be used in combination with any embedding algorithm that operates on distance matrices.

### 3.1 THE PROBLEM – INFORMALLY

Given a set of $n$ samples $X$ from a high dimensional space, e.g. $\mathbb{R}^d$, our goal is to find a low dimensional representation $Y$ in $\mathbb{R}^2$ that captures the local structure in $X$ while factoring out prior

knowledge $Z$ about the samples. Here, we consider both high dimensional data $X$ and prior $Z$ to be given as distance matrices, thus allowing for data from typical spaces such as Euclidean, but also images, up to unstructured data such as texts or graphs, for which distance matrices can be specified using a kernel. When embedding $X$, our goal is to embed the samples such that the pairwise low dimensional distances $D^Y$ resemble high dimensional distances $D^X$ locally, but are distinct to the prior distances $D^Z$. Informally, we can state this goal as finding an embedding $Y$ subject to

$$D^X \approx D^Y \not\approx D^Z \ .$$

We could formally define this as a multi-objective problem composed of a minimization over the difference between $D^X$ and $D^Y$ and a maximization of the difference between $D^Y$ and $D^Z$. Besides how to measure these differences, there are two problems that render classic multi-objective optimization impractical. First, the two functions are highly imbalanced, with the minimization objective obtaining its optimum at $0$ and the maximization at $+\infty$, hence we need to constrain the optimization. Second, we want to put emphasis on correctly reconstructing local structure, as this yields superior visualizations (van der Maaten & Hinton, 2008; McInnes et al., 2018).

### 3.2 THE PROBLEM – INFORMATION THEORETICALLY

The t-distributed Stochastic Neighbor Embedding (tSNE) (van der Maaten & Hinton, 2008) is a state-of-the-art approach for embedding data into low dimensional spaces that preserves the local structure of the high dimensional data. In particular, it models the local neighborhood of a point by casting the pairwise distances into similarity distributions that express for each point $i$ the likelihood of observing point $j$ as neighbor, given by $p_{j|i}$. For the high dimensional distances $D_{ij}^X$, this likelihood is approximated by a Gaussian kernel centered at point $i$

$$p_{j|i} = \frac{\exp(-(D_{ij}^X)^2/2\sigma_i^2)}{\sum_{k\neq i}\exp(-(D_{ik}^X)^2/2\sigma_i^2)}.$$

To account for varying densities of points in the space, the variance $\sigma_i$ is dependent on where the kernel is centered. Given the user specified parameter *perplexity*, which can be thought of as an estimate of the neighborhood size, we can solve $perplexity = 2^{H(P_i)}$ for $\sigma_i$ for each point $i$, where $H(P_i) = \sum_j p_{j|i} \log p_{j|i}$ is the entropy. By symmetrizing the conditional probabilities, the joint probability of a pair of points is given as $p_{ij} = \frac{p_{j|i}+p_{i|j}}{2n}$, which yields the desired local similarity representation of high dimensional points.

The low dimensional point similarities $q_{ij}$ are represented by a t-distribution instead of a Gaussian, which solves the *crowding problem*[1] due to its heavy tails. We thus get low dimensional similarities

$$q_{ij} = \frac{(1+(D_{ij}^Y)^2)^{-1}}{\sum_{k\neq l}(1+(D_{kl}^Y)^2)^{-1}}.$$

The goal of tSNE is to model pairs of points exhibiting a high similarity in the high dimensional space to have a high similarity in the low dimensional space. This is achieved by minimizing the Kullback-Leibler Divergence (KL), given by $D_{\mathrm{KL}}(P \mid\mid Q) = \sum_{i\neq j} p_{ij} \log \frac{p_{ij}}{q_{ij}}$, for the pairwise probabilities. This information theoretic measure yields the number of excess bits needed if we would encode $P$ using a code optimal for encoding $Q$ and thus models how well $Q$ approximates $P$. Minimizing the KL divergence with respect to $Y$, we get a non-convex objective that we can practically optimize using gradient descent. Using a similar notion of neighborhood distributions, we can now define a new objective that instantiates our objective using tools from information theory.

### 3.2.1 FACTORING OUT PRIOR INFORMATION WITH JEDI

To incorporate prior information into the tSNE objective, we first need to model the neighborhood distribution $P'$ of the prior. Similar to the high dimensional data, we use a Gaussian kernel by which we hence put emphasis on samples that are close according to the prior, defined as

$$p'_{j|i} = \frac{\exp(-(D_{ij}^Z)^2/2\sigma_i^2)}{\sum_{k\neq i}\exp(-(D_{ik}^Z)^2/2\sigma_i^2)},$$

---

[1]The crowding problem is the phenomenon of assembling all points in the center of the map, due to the accumulation of many small attractive forces as moderate distances are not accurately modelled.

with $\sigma_i$ a perplexity parameter which describes the desired neighborhood size in the prior space.

We thus search for a similarity distribution $Q$ of points $Y$ in the low dimensional space, that is similar to the high dimensional similarities $P$ but different from the prior similarities $P'$. Similar to tSNE, the first term of our objective corresponds to minimizing the KL divergence, thus a natural extension would be to add a second term that rewards maximizing the KL divergence between the distances of embedding $Q$ and prior $P'$. This would be naive, however, as this second term would dominate the optimization because KL divergence is unbounded. Furthermore, it would not allow us to exploit the asymmetry of divergence in the one or the other direction.

To mitigate the unboundedness, the skewed KL divergence has been proposed, mixing the two distributions $D_{\mathrm{KL}}^\beta(P \,||\, Q) = D_{\mathrm{KL}}(P \,||\, (1-\beta)P + \beta Q)$ with $\beta \in [0,1]$ controlling skewness and thus boundedness (see e.g. Yamano (2019)). To obtain symmetry, the $\beta$-Jensen-Shannon divergence defined as $\mathrm{JS}_\beta(P \,||\, Q) = \frac{1}{2}(D_{\mathrm{KL}}^\beta(P \,||\, Q) + D_{\mathrm{KL}}^\beta(Q \,||\, P))$ was introduced. Based on these ideas, we propose a new divergence, which we call parameterized Jensen-Shannon Divergence (pJSD), which allows to control for both, the level of skewness as well as the level of symmetry, and prove that pJSD is bounded.

**Definition 1** (Parameterized Jensen Shannon Divergence)**.** *For two probability distributions $P'$ and $Q$ we define the parameterized Jensen-Shannon divergence as*

$$\mathrm{JS}_\beta^\alpha(P' \,||\, Q) = \alpha D_{\mathrm{KL}}(P' \,||\, \beta Q + (1-\beta)P') + (1-\alpha)D_{\mathrm{KL}}(Q \,||\, \beta P' + (1-\beta)Q),$$

*where $0 \leq \alpha \leq 1$ determines the level of symmetry and $0 < \beta < 1$ determines the level of skewness.*

**Theorem 1** (Upper bound on pJSD)**.** *For $0 \leq \alpha \leq 1$ and $0 < \beta < 1$ the parametrized JS divergence is bounded from above by*

$$\mathrm{JS}_\beta^\alpha \leq -\log(1-\beta).$$

We provide a proof in App. A.1.2. Putting the pieces together, we can now formulate our objective as the minimization of the KL divergence between the similarity distributions of $X$ and $Y$, and the maximization of the pJSD between the similarity distributions of $Y$ and $Z$, which is

$$\arg\min_Y D_{\mathrm{KL}}(P \,||\, Q) - \mathrm{JS}_\beta^\alpha(P' \,||\, Q).$$

While the parameters $\alpha, \beta$ give the user flexibility on how much of the prior distribution should be factored out, we will later discuss a good default parameter instantiation based on synthetic data. This objective, similar to vanilla tSNE and related approaches such as ctSNE, is not convex, and is optimized using gradient descent. We provide the derivation of the gradients in App. A.1.1. This provides us with a method that solves our problem of factoring out prior knowledge on information theoretic grounds, which we call JEDI in resemblance of the *Je*nsen Shannon *Di*vergence.

**Computational Complexity**   The computational and memory complexity of JEDI is in $O(kn^2)$, for $n$ samples and $k$ iterations, which comes from the summation over all pairs of sample in the divergences in each iteration. Due to the interactions in the gradient of pJSD, standard algorithmic optimizations of tSNE (van der Maaten, 2014; Linderman et al., 2019) are not directly applicable.

Overall, JEDI is a powerful, theoretically appealing approach to factor out prior knowledge. Based on tSNE, JEDI inherits many of its strengths and weaknesses. In particular runtime and strong emphasis on local structure make it hard to successfully apply tSNE, and therewith JEDI, to datasets that are either very large and/or contain structure at different scales, i.e. data as typically considered in bioinformatics. In the next section, we therefore revisit the problem, and propose an embedding algorithm independent approach that is applicable to such settings.

## 3.3   THE PROBLEM – ALGORITHM INDEPENDENTLY

UMAP is one of the state-of-the-art competitors to tSNE that alleviates its drawbacks for large data that not only contain local structure. Rather than presenting a dedicated solution for UMAP, we here propose a general, embedding-formulation independent approach. To do so, we have to revisit the original problem formulation, where the goal is to approximate high dimensional distances with embedding distances while simultaneously keeping them far from the prior. The key idea here is that, if we factor out the prior knowledge from the high dimensional distances directly, we are

independent of the actual embedding process, and hence any embedding algorithm can be used. Informally, we can state this as $(D^X \ominus D^Z) \approx D^Y$, where $\ominus$ describes some yet to be defined way of factoring out prior knowledge $Z$ from the distances over high dimensional data $X$. Once we have this operator, we can use any distance metric based embedding algorithm on its result to obtain high quality embeddings $Y$ from which $Z$ has been factored out. Clearly, the operator should result in a proper distance metric, discard any structure that is evident and keep any structure that is not evident given prior knowledge $Z$. W.l.o.g., for the remainder of this section we assume that the distances $D$ are scaled to $D' = \frac{1}{D_{max}}D$, with $D_{max}$ the maximum value in $D$. We define operator $\ominus$ as

$$(D^X \ominus_\lambda D^Z)_{ij} = \begin{cases} D_{ij}^X - \frac{1}{2}\lambda D_{ij}^Z + \lambda & i \neq j \ , \\ D_{ij}^X & i = j \ , \end{cases}$$

which is to say, we subtract the information given by the prior distances from the high dimensional distances in a linear form, with $\lambda$ controlling how much prior to be removed. Although surprisingly simple, this elegant definition has very convenient properties that render it very powerful. First, there is only a single parameter $\lambda$, which due to linearity gives the user direct and interpretable control over how much the prior information should be taken into account. Second, the distance matrix we obtain by applying the operator maintains metric properties – the proof can be found in App. A.1.3 – that are required to properly optimize downstream embedding algorithms. Without the guarantee of symmetry and triangle inequality optimizing over pairwise distances is not possible.

**Theorem 2** (Metric). *Assuming that $D^X$ and $D^Z$ are based on valid metrics, for any $\lambda > 0$, $(D^X \ominus_\lambda D^Z)$ fulfills the metric axioms of non-negativity, symmetry, identity, and triangle inequality.*

Furthermore, the operator has the property of maintaining the original structure under an uninformative prior. More formally, we define $N_k^D(i) = \{j \in kNN(i) \text{ according to } D\}$ to be the $k$-neighborhood of sample $i$ according to distances $D_{ij}$. For simplicity of notation, we will assume that all distances are distinct, the results and definitions can directly be generalized to the case of equal distances. Assuming an uninformative prior, which was generated independently of the high dimensional data $D^Z \perp\!\!\!\perp D^X$, on expectation the neighborhoods of each point stay the same.

**Theorem 3** (Uninformative prior (proof in App. A.1.4)). *Assume the prior is uninformative, that is $D^Z \perp\!\!\!\perp D^X$. Furthermore, the distances are normalized to $D_{ij}^Z \in [0, 1]$. For fixed $\lambda > 0$ let $(F_\lambda)_{ij} = D^X \ominus_\lambda D^Z$. On expectation, the neighborhoods in $X$ and in $X$ with factored out prior are the same, that is $\forall i, k. N_k^{F_\lambda}(i) =_{E[.]} N_k^{D^X}(i)$.*

This theorem proves that the embeddings obtained from this distance matrix are expected to be robust against priors unrelated to the input, and thus no knowledge is lost and no spurious knowlegde is generated. Both the metricity as well as the robustness against uninformative priors are essential for generating good embeddings, which more complex formulations can not provide that easily. Normalizing the distances as discussed above, and applying the $\ominus_\lambda$ to factor out prior knowledge, we obtain an embedding algorithm independent method to factor out prior knowledge. In reminiscence of how the plots look, we refer to this method as CONFETTI and give its pseudocode as Alg. 1.

**Complexity**   CONFETTI runs in time $O(n^2)$, which includes the normalization of the distance matrix and computation of the operator, which only counts towards a very small constant. Additionally, we will need to run an embedding algorithm, which respective runtime is added to $O(n^2)$.

## 4   EXPERIMENTS

We evaluate on both synthetic and real world data. We make the implementations of JEDI and CONFETTI available online.[2] Since there does not exist any direct competitor that can factor out arbitrary distance matrices from an embedding, we compare to two closest competitors. The first is ctSNE (Kang et al., 2019), which extends the tSNE objective and can factor out prior information given as cluster labels. The second is supervised LLE (De Ridder et al., 2003), which although originally designed to emphasize structure in the embedding given as labels, we can modify such that it instead emphasizes any structure *not* in the prior. We give the details for this modification, which we refer to as sLLE$^{-1}$, in App. B.1. For fair comparison, we optimize all parameters via grid search on a synthetic data hold out set, and use these throughout all experiments (see App. B.2).

---

[2]Available after publication.

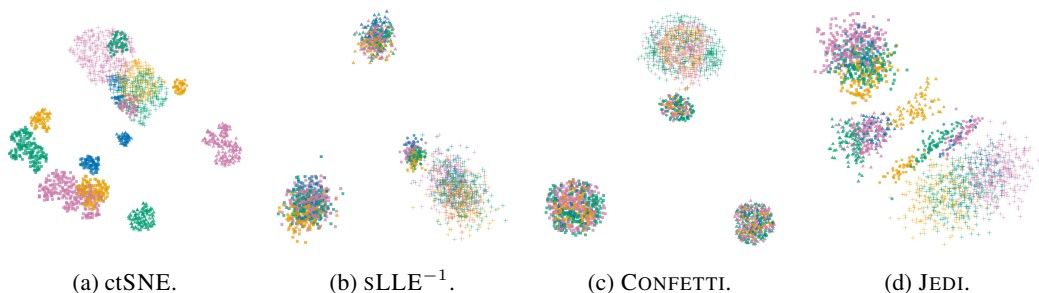

(a) ctSNE.   (b) sLLE$^{-1}$.   (c) CONFETTI.   (d) JEDI.

Figure 1: *Synthetic data.* Shown are conditional embeddings of synthetic data given resp. ground truth labels for ctSNE and sLLE$^{-1}$ (a,b), and euclidean distance for CONFETTI and JEDI (c,d). Colors correspond to cluster assignment over dimensions 1–8, shape (circle, square, triangle, and cross) to cluster assignment over dimensions 9–12. In b),c), and d), each of the four clusters consists of samples of one shape.

## 4.1 RELIABLY FACTORING OUT DISTANCE PRIORS

We first consider synthetic data with known ground truth. In particular, we consider synthetic data of $n = 2\,000$ samples over 14 dimensions, where dimensions 1-8 and 9-12 both exhibit 4 clusters of different sizes, while dimensions 13 and 14 are Gaussian noise. We give more details, as well as a tSNE plot in App. B.3. The cluster structure over the first 8 dimensions dominates the tSNE embedding. To discover information beyond these clusters, we provide JEDI and CONFETTI the euclidean distances over these 8 dimensions. To enable a fair comparison to ctSNE and sLLE$^{-1}$, which are limited to label priors, we provide the *ground truth* cluster assignment as background knowledge. All methods finished within minutes, and we plot the results in Fig. 1. Although given the true labelling, ctSNE fails to satisfactorily factor out the prior knowledge, whereas our methods yield the 4 distinct clusters from dimension 9-12. Notably, when we provide JEDI with the ground truth label assignment, it yields similarly sharp clusters as sLLE$^{-1}$ (see App. 13).

To objectively quantify how well prior knowledge is factored out from an embedding, we propose to measure the similarity over neighborhoods. For two distance matrices $D, D'$ and neighborhood size $k$, we define the neighbourhood overlap score (NOS) as $\text{NOS}(D, D', k) = \frac{1}{n}\frac{1}{k}\sum_{i=1}^{n} |\{kNN \text{ of } i \text{ in } D\} \cap \{kNN \text{ of } i \text{ in } D'\}|$. Correspondingly, for a distance matrix $D$ and label distribution $L$, we get $\text{NOS}(D, L, k) = \frac{1}{n}\frac{1}{k}\sum_{i=1}^{n} \frac{1}{|L_i|}|\{kNN \text{ of } i \text{ in } D\} \cap \{L_i\}|$. While this score lends itself for evaluation, it is hard to directly optimize (see App. B.3). Plotting $\text{NOS}(D^X, D^Y, )$ for all neighborhood sizes $k = 1 \ldots n$ allows us to asses how well we preserve information of the original data, whereas plotting $\text{NOS}(D^X, D^Z)$ allows us to asses how well we factor out prior knowledge from an embedding. As the id-line corresponds to a random neighbor encounter, we can measure the area between the NOS curve and id-line as a proxy for how well we preserve information, respectively how well we factor out prior knowledge.

For the synthetic data, the area between the curves and the id-line for ctSNE, sLLE$^{-1}$, JEDI, and CONFETTI are .237, .344, .340, .344 when we compute the NOS between embedding and input data without prior. For the NOS between embedding and prior labels we have .037, .005, .003, .022, respectively (plots are given in App. Fig. 14). We see that JEDI factors out the prior almost ideally, as well as sLLE$^{-1}$, and that CONFETTI performs slightly worse especially in small neighborhoods. When evaluating the NOS with regard to the euclidean distance prior, CONFETTI beats all other methods with an area between curves of only 0.002. Overall, ctSNE shows the worst NOS performance, which is also evident in the embedding (see Fig. 1). As for information captured in the embedding from the non-prior dimensions, JEDI, CONFETTI, and sLLE$^{-1}$ do equally well, putting ctSNE at a distance. Overall, JEDI and CONFETTI are perform on par with the state of the art given label priors, but perform at least as well given continuous priors, for which ctSNE and sLLE$^{-1}$ are not applicable.

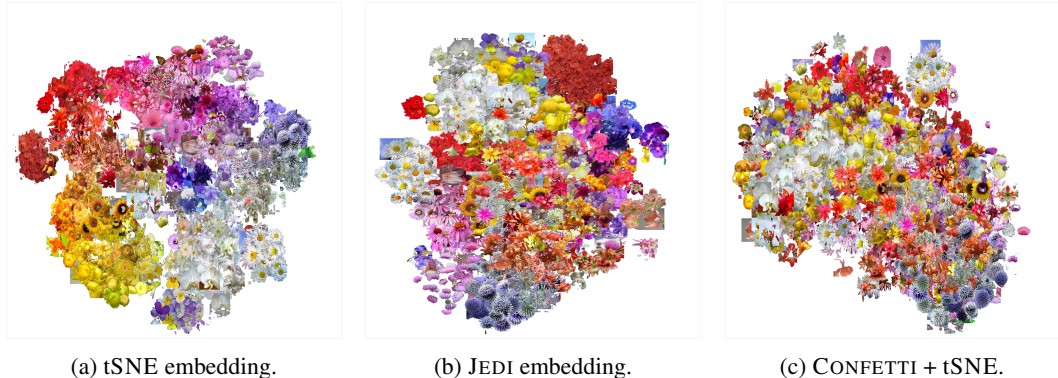

| (a) tSNE embedding. | (b) JEDI embedding. | (c) CONFETTI + tSNE. |

Figure 2: *Real world data*. Embedded are the sum of chi-square matrices for color, local shape and texture, boundary shape, and spatial petal distribution of the Oxford flower data. Shown are vanilla tSNE (left), and JEDI (middle) and CONFETTI (right) with HSV color distances as prior knowledge.

## 4.2 RECOVERING FLOWER GEOMETRY

To evaluate on real data, we consider the Oxford flower dataset from Nilsback & Zisserman (2008), which consists of over 8000 images of flowers of 102 different classes. The data is given as a set of four pairwise distance matrices which are the Chi-squared distances of the color (HSV), the local shape and texture, the shape of the boundary, and the spatial distribution of petals of the flower, all computed on segmented flower images. The available implementations for ctSNE and $sLLE^{-1}$ are not applicable to distance matrices as input $X$, and hence we could not compare to these methods. We will use the sum of all four matrices as high dimensional input. To keep the results interpretable, we subsample 40 images from 25 different classes each, by which we have $n = 1000$ samples. We are interested whether our algorithms is able to factor out a known prior that dominates an embedding in a real world setting, which is why we specify the HSV color matrix as background knowledge. While other priors could be specified, we would not know if that prior would be present in the input data and thus would not be able to evaluate success. CONFETTI and JEDI terminate in seconds, respectively two minutes.

We give the vanilla tSNE embedding in Fig. 2a, which besides a clustering according to colour conveys little other information. When we factor out the color information with JEDI and CONFETTI, we see that colors mix and new clusters form according to other features. For example, spiky petals arrange at the one side (Fig. 2b bottom, Fig. 2c bottom right), whereas rounded petals assemble on the respective other side of the space. Similarly, flowers with few but large petals gather on one side (Fig. 2b right, Fig. 2c top) and flowers with many thin petals on the respective other side. Apart from these visual changes, we can evaluate based on the NOS plot, which shows that also for this metric prior our methods are able to factor out the background knowledge (see App. Fig. 15).

## 4.3 BATCH EFFECTS IN SINGLE CELL SEQUENCING

One of the major applications of embeddings such as tSNE and UMAP is in single cell sequencing, where it is a standard routine to visualize the sequencing data, allowing e.g. to assess the quality of sequencing, to easily remove outliers from the data, or highlight differences between cohorts. To test the methods on such data, we use a recent single cell data set of cerebrospinal fluid (CSF) and whole blood of multiple sclerosis patients and a control group (Schafflick et al., 2020). We generate a vanilla UMAP embedding using their original jupyter notebooks, and plot it as Fig. 3a. It shows the typical clustering according to the cell types in blood and CSF, when coloring the samples according to gene expression of standard marker genes (for more information refer to the original paper). Here, we are interested whether our methods can reveal information beyond cell type. We thus provide again the gene expression as input, and additionally take the euclidean distance between the marker gene expression levels of each sample as prior, corresponding to how the coloring was obtained. As ctSNE cannot deal with continuous priors, we instead provide it the cluster labels from an agglomerative clustering of the marker gene expression levels, which yielded the same number

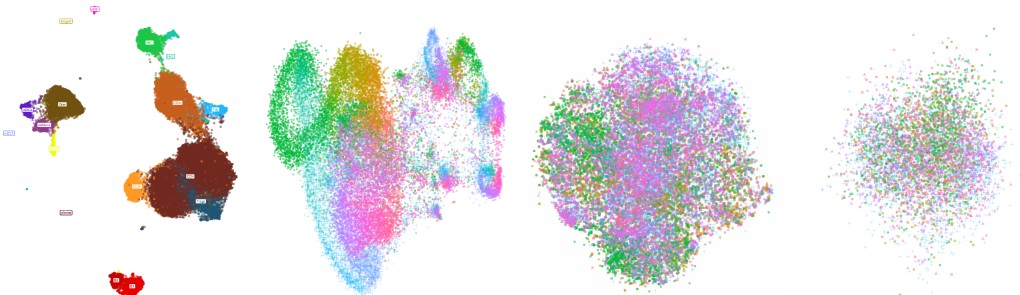

(a) UMAP embedding, coloring according to cell type.

(b) CONFETTI + UMAP with marker gene expression as prior.

(c) ctSNE with labels of clustering marker gene expression as prior.

(d) sLLE$^{-1}$ with labels of clustering marker gene expression as prior.

Figure 3: *Real World Data* Visualizations of the embedding of single cell sequencing data of UMAP, CONFETTI + UMAP, ctSNE, and sLLE$^{-1}$. sLLE$^{-1}$ failed to produce an embedding for the full data, results are obtained for a random subset of the data. Coloring in b), c), and d) according to batch ID, shape according to sample type (case vs control, and blood vs CSF).

of clusters as the original paper (see App. Fig. 16b). sLLE$^{-1}$ requires a matrix inversion, and for this data gives an error hinting that the data matrix is too large. Hence, we provide sLLE$^{-1}$ with only a random subset of ~6000 samples to obtain an embedding. While this is not practical, as important tasks such as outlier detection or assignment of cell types or states can only be done for that particular subset, it allows us to compare the obtained embeddings. We provide sLLE$^{-1}$ with the same cluster labels as for ctSNE. ctSNE terminated within 23 minutes, sLLE$^{-1}$ in 32 minutes, CONFETTI in 10 minutes, and JEDI in 100h, the resulting embeddings are given in Figure 3, a vanilla UMAP plot with coloring according to batch ID is given in App. Fig. 17c.

The results of CONFETTI show a surprising separation of samples from different batches and accordingly separation of the different tissue (blood and CSF), and of case and control along a manifold. Encouraged by these results, we also apply CONFETTI with tissue information as prior, and then observe that the previously separated clusters for actually equal cell types (B1,B2, NK1,NK2, mDC1,mDC2) are now merged (see App. Fig. 17a), while all other information is kept. For ctSNE, there are no clusters or manifolds directly visible and the result is a rather homogeneous ball, which show only a clear separation of CD4 cells, which make up the largest proportion of cells (compare App. Fig. 16a). Similar to ctSNE, JEDI shows no clear separation of cell types (see App. Fig. 17b), but there is no bias towards CD4 cells as for ctSNE. sLLE$^{-1}$ does not show any discernable structure beyond the provided cell types.

## 5 DISCUSSION AND CONCLUSION

In the experiments, we show that both JEDI and CONFETTI correctly factor out prior knowledge and result embeddings that reveal previously hidden structure. On synthetic data, when provided with euclidean distances capturing the structure which dominates the vanilla tSNE embedding, both our methods reveal the clustering of the input data that is independent of the background knowledge. Our closest competitor ctSNE, is not able to factor out the background knowledge well, and its embeddings still show structure of the prior even when given the ground truth cluster labels. When we provide our methods the same information, they do recover the correct clustering, demonstrating their ability to reliably factor out both distances as well as label priors.

On real world data of flower images, where only distance matrices between attributes of the images are available, both JEDI and CONFETTI are able to factor out the property dominating the vanilla tSNE embedding, and organizing flowers according to number and shape of petals rather than by color. This shows that both are able applicable to real world settings, and unlike their competitors, can consider both input data and background knowledge in the form of different distance metrics. This provides us with the advantage to be applicable in domains where the input is only available as distance matrices, e.g. kernel-derived distances for unstructured data such as graphs or strings.

On the perhaps most widespread application of low dimensional embeddings, single cell gene expression data, we confirm that tSNE based approaches are a bad fit; both ctSNE and JEDI get lost in local detail, fail to factor out the prior knowledge, and are slow. When we combine our algorithm-independent approach, CONFETTI, with UMAP we do arrive at meaningful embeddings from which the background knowledge has been factored out. In particular, when we provide it with prior knowledge of marker gene expression – which is used to determine cell type – we arrive at an embedding that organizes samples according to batch id and tissue type. Conversely, if provided with tissue type as prior, we observe that previously separate clusters, that actually contain cells of the same type, are merged.

In practice, a distance based prior is usually the most natural representation of the background knowledge that cannot be cast easily into class labels. With binning or clustering using distances, information is lost for differences *within* a class, and the relative information is lost *between* classes. For example, effects based on geographic locations, age differences, or income that should be factored out from an embedding should be considered as is rather than cast into labels, which is not possible with current approaches. Our approaches do take into account these relative relationships by considering distances between pairs of samples, such as the geodesic distance between locations of two individuals or their difference in age, which would be lost when encoded by labels.

Overall, both our proposals solve the problem of factoring prior knowledge from low dimensional embeddings, and by allowing the prior knowledge to be specified as distances they are both applicable to a large number of data types and domains. While JEDI has a theoretically appealing objective, it inherits all strength and weaknesses of tSNE by which especially its runtime is an open problem. For tSNE, several methods have been proposed that significantly speed up the optimization, yet none of them seem directly applicable to JEDI. We leave their adaptation for future work. Furthermore, although yielding overall good results, we would like to more closely investigate the parameters of tSNE and JEDI in a more principled way, as for new domains parameter settings for e.g. tSNE are mostly found by trial and error. Here, we tested our methods on different types distance metrics that cover different applications, and with single cell gene expressions one of the most important domains for low dimensional embeddings. In the future, we would also like to investigate how our methods perform when provided with more exotic types of background knowledge.

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

# A  METHOD APPENDIX

## A.1  THEORETICAL RESULTS

### A.1.1  WORKING TITLE: THE HIGH GROUND

In the following, we derive the gradient of the conditional cost function

$$C(Y) = D_{\mathrm{KL}}(P \,\|\, Q) - \mathrm{JS}_\beta^\alpha(P' \,\|\, Q).$$

We split the gradient into two parts

$$\frac{\delta C}{\delta y_i} = \frac{\delta D_{\mathrm{KL}}(P \,\|\, Q)}{\delta y_i} - \frac{\delta \mathrm{JS}_\beta^\alpha(P' \,\|\, Q)}{\delta y_i},$$

as the first is given by the tSNE gradient with

$$\frac{\delta D_{\mathrm{KL}}(P \,||\, Q)}{\delta y_i} = 4 \sum_j (p_{ij} - q_{ij}) \left(1 + \|y_i - y_j\|^2\right)^{-1} (y_i - y_j).$$

First, we write out the full parametrized Jensen-Shannon Divergence:

$$\begin{aligned}
\mathrm{JS}_\beta^\alpha(P' \,||\, Q) &= \alpha D_{\mathrm{KL}}(P' \,||\, \beta Q + (1-\beta)P') + (1-\alpha)D_{\mathrm{KL}}(Q \,||\, \beta P' + (1-\beta)Q) \\
&= \alpha \sum_{i,j} p'_{ij} \log \frac{p'_{ij}}{\beta q_{ij} + (1-\beta)p_{ij}} + (1-\alpha) \sum_{i,j} q_{ij} \log \frac{q_{ij}}{\beta p'_{ij} + (1-\beta)q_{ij}} \\
&= \alpha \sum_{i,j} p'_{ij} \log p'_{ij} - p'_{ij} \log \left(\beta q_{ij} + (1-\beta)p_{ij}\right) \\
&\quad + (1-\alpha) \sum_{i,j} q_{ij} \log q_{ij} - q_{ij} \log \left(\beta p'_{ij} + (1-\beta)q_{ij}\right)
\end{aligned}$$

The expected similarities $p'_{ij}$ are symmetrized probabilities

$$p'_{ij} = \frac{p'_{j|i} + p'_{i|j}}{2n} \qquad \text{with} \qquad p'_{j|i} = \frac{\exp(-(D_{ij}^Z)^2/2(\sigma'_i)^2)}{\sum_{k \neq i} \exp(-(D_{ik}^Z)^2/2(\sigma'_i)^2)}$$

and the low-dimensional similarities $q_{ij}$ are defined as

$$q_{ij} = \frac{(1 + \|y_i - y_j\|^2)^{-1}}{\sum_{k \neq l}(1 + \|y_k - y_l\|^2)^{-1}},$$

where we abbreviate the distances by $d_{ij} = \|y_i - y_j\|$ and the normalization term by $Z = \sum_{k \neq l}(1 + (D_{kl}^Y)^2)^{-1}$, leading to the succinct notation $q_{ij} = (1 + (D_{ij}^Y)^2)^{-1}Z^{-1}$.

As $y_i$ only affects the distances $D_{ij}^Y$ and $D_{ji}^Y$ by

$$\frac{\delta d_{ij}}{\delta y_i} = \frac{\delta \|y_i - y_j\|}{\delta y_i} = \frac{y_i - y_j}{D_{ij}^Y},$$

we will derive the gradient with respect to $D_{ij}^Y$

$$\begin{aligned}
\frac{\delta C}{\delta y_i} &= \sum_j \left( \frac{\delta C}{\delta D_{ij}^Y} \frac{\delta D_{ij}^Y}{y_i} + \frac{\delta C}{\delta D_{ji}^Y} \frac{\delta D_{ji}^Y}{\delta y_i} \right) \\
&= 2 \sum_j \frac{\delta C}{\delta D_{ij}^Y} \frac{y_i - y_j}{D_{ij}^Y}.
\end{aligned}$$

Note that the first term of the Jensen-Shannon cost function is constant with respect to $D_{ij}^Y$. Thus, we split the gradient

$$\begin{aligned}
\frac{\delta \mathrm{JS}_\beta^\alpha(P' \,||\, Q)}{\delta D_{ij}^Y} &= -\alpha \sum_{k \neq l} \frac{\delta p'_{kl} \log \left(\beta q_{kl} + (1-\beta)p_{kl}\right)}{\delta D_{ij}^Y} \\
&\quad + (1-\alpha) \sum_{k \neq l} \frac{\delta q_{kl} \log q_{kl}}{\delta D_{ij}^Y} \\
&\quad - (1-\alpha) \sum_{k \neq l} \frac{\delta q_{kl} \log \left(\beta p'_{kl} + (1-\beta)q_{kl}\right)}{\delta D_{ij}^Y},
\end{aligned}$$

and compute the three parts separately. The building block are the derivatives of $Z$ and $q_{kl}$ with respect to $D_{ij}^Y$:

$$
\begin{aligned}
\frac{\delta Z}{\delta D_{ij}^Y} &= \sum_{k \neq l} \frac{\delta(1 + (D_{kl}^Y)^2)^{-1}}{\delta D_{ij}^Y} \\
&= \frac{\delta(1 + (D_{ij}^Y)^2)^{-1}}{\delta D_{ij}^Y} \\
&= (-1)(1 + (D_{ij}^Y)^2)^{-2} 2 D_{ij}^Y \\
&= -2 D_{ij}^Y (1 + (D_{ij}^Y)^2)^{-2} \\
&= -2 D_{ij}^Y q_{ij} Z (1 + (D_{ij}^Y)^2)^{-1}
\end{aligned}
$$

$$
\begin{aligned}
\frac{\delta q_{kl}}{\delta D_{ij}^Y} &= \frac{\delta \frac{(1+(D_{ij}^Y)^2)^{-1}}{Z}}{\delta D_{ij}^Y} \\
&= \frac{1}{Z^2} \left( \frac{\delta(1 + (D_{kl}^Y)^2)^{-1}}{\delta D_{ij}^Y} Z - (1 + (D_{kl}^Y)^2)^{-1} \frac{\delta Z}{\delta D_{ij}^Y} \right) \\
&= \frac{1}{Z^2} \left( \frac{\delta(1 + (D_{kl}^Y)^2)^{-1}}{\delta D_{ij}^Y} Z - (1 + (D_{kl}^Y)^2)^{-1} \left( -2 D_{ij}^Y (1 + (D_{ij}^Y)^2)^{-2} \right) \right) \\
&= \frac{1}{Z^2} \left( \frac{\delta(1 + (D_{kl}^Y)^2)^{-1}}{\delta D_{ij}^Y} Z + (1 + (D_{kl}^Y)^2)^{-1} 2 D_{ij}^Y (1 + (D_{ij}^Y)^2)^{-2} \right) \\
&= \frac{1}{Z} \frac{\delta((1 + (D_{kl}^Y)^2)^{-1})}{\delta D_{ij}^Y} + (1 + (D_{kl}^Y)^2)^{-1} 2 D_{ij}^Y \frac{(1 + d_{ij}^2)^{-2}}{Z^2} \\
&= \frac{\delta((1 + (D_{kl}^Y)^2)^{-1})}{\delta D_{ij}^Y} \frac{1}{Z} + (1 + (D_{kl}^Y)^2)^{-1} 2 D_{ij}^Y q_{ij}^2 \\
&= \begin{cases} (1 + (D_{ij}^Y)^2)^{-1} 2 D_{ij}^Y q_{ij}(q_{ij} - 1) & \text{if } kl = ij \\ (1 + (D_{kl}^Y)^2)^{-1} 2 D_{ij}^Y q_{ij}^2 & \text{if } kl \neq ij. \end{cases}
\end{aligned}
$$

Next we derive the gradients of the three parts:

$$
\begin{aligned}
\sum_{k \neq l} \frac{\delta p'_{kl} \log \left(\beta q_{kl} + (1-\beta)p'_{kl}\right)}{\delta d_{ij}} &= \sum_{k \neq l} p'_{kl} \frac{\delta \log \left(\beta q_{kl} + (1-\beta)p'_{kl}\right)}{\delta d_{ij}} \\
&= \sum_{k \neq l} \frac{\beta p'_{kl}}{\beta q_{kl} + (1-\beta)p'_{kl}} \frac{\delta q_{kl}}{\delta d_{ij}} \\
&= -\frac{\beta p'_{ij}(1+d_{ij}^2)^{-1}2d_{ij}q_{ij}}{(\beta q_{ij} + (1-\beta)p'_{ij})} + \sum_{k \neq l} \frac{\beta p'_{kl}(1+d_{kl}^2)^{-1}2d_{ij}q_{ij}^2}{\beta q_{kl} + (1-\beta)p'_{kl}} \\
&= 2d_{ij}\left(-\frac{\beta p'_{ij}q_{ij}(1+d_{ij}^2)^{-1}}{\beta q_{ij} + (1-\beta)p'_{ij}} + q_{ij}^2 \sum_{k \neq l} \frac{\beta p'_{kl}(1+d_{kl}^2)^{-1}}{\beta q_{kl} + (1-\beta)p'_{kl}}\right) \\
&= 2d_{ij}\left(-\frac{\beta p'_{ij}q_{ij}(1+d_{ij}^2)^{-1}}{\beta q_{ij} + (1-\beta)p'_{ij}} + q_{ij}^2 \sum_{k \neq l} \frac{\beta p'_{kl}q_{kl}Z}{\beta q_{kl} + (1-\beta)p'_{kl}}\right) \\
&= 2d_{ij}q_{ij}(1+d_{ij}^2)^{-1}\left(-\frac{\beta p'_{ij}}{\beta q_{ij} + (1-\beta)p'_{ij}} + \sum_{k \neq l} \frac{\beta p'_{kl}q_{kl}}{\beta q_{kl} + (1-\beta)p'_{kl}}\right)
\end{aligned}
$$

$$
\begin{aligned}
\sum_{k \neq l} \frac{\delta q_{kl} \log q_{kl}}{\delta D_{ij}^Y} &= \sum_{k \neq l} q_{kl} \frac{\delta(\log q_{kl})}{\delta D_{ij}^Y} + \frac{\delta q_{kl}}{\delta D_{ij}^Y} \log q_{kl} \\
&= \sum_{k \neq l} q_{kl} \frac{1}{q_{kl}} \frac{\delta q_{kl}}{\delta D_{ij}^Y} + \frac{\delta q_{kl}}{\delta D_{ij}^Y} \log q_{kl} \\
&= \sum_{k \neq l} \frac{\delta q_{kl}}{\delta D_{ij}^Y}(1 + \log q_{kl}) \\
&= -(1+(D_{ij}^Y)^2)^{-1}2D_{ij}^Y q_{ij}(1 + \log q_{ij}) + \sum_{k \neq l}(1+(D_{kl}^Y)^2)^{-1}2D_{ij}^Y q_{ij}^2(1 + \log q_{kl}) \\
&= 2D_{ij}^Y\left(-(1+(D_{ij}^Y)^2)^{-1}q_{ij}(1 + \log q_{ij}) + q_{ij}^2 \sum_{k \neq l}(1+(D_{kl}^Y)^2)^{-1}(1 + \log q_{kl})\right) \\
&= 2D_{ij}^Y q_{ij}(1+(D_{ij}^Y)^2)^{-1}\left(-(1 + \log q_{ij}) + \sum_{k \neq l} q_{kl}(1 + \log q_{kl})\right)
\end{aligned}
$$

$$\sum_{k \neq l} \frac{\delta q_{kl} \log \left( \beta p'_{kl} + (1 - \beta) q_{kl} \right)}{\delta D^Y_{ij}}$$

$$= \sum_{k \neq l} q_{kl} \frac{\delta \log \left( \beta p'_{kl} + (1 - \beta) q_{kl} \right)}{\delta D^Y_{ij}} + \frac{\delta q_{kl}}{\delta D^Y_{ij}} \log \left( \beta p'_{kl} + (1 - \beta) q_{kl} \right)$$

$$= \sum_{k \neq l} \frac{(1 - \beta) q_{kl}}{\beta p'_{kl} + (1 - \beta) q_{kl}} \frac{\delta q_{kl}}{\delta D^Y_{ij}} + \frac{\delta q_{kl}}{\delta D^Y_{ij}} \log \left( \beta p'_{kl} + (1 - \beta) q_{kl} \right)$$

$$= \sum_{k \neq l} \frac{\delta q_{kl}}{\delta D^Y_{ij}} \left( \frac{(1 - \beta) q_{kl}}{\beta p'_{kl} + (1 - \beta) q_{kl}} + \log \left( \beta p'_{kl} + (1 - \beta) q_{kl} \right) \right)$$

$$= -(1 + (D^Y_{ij})^2)^{-1} 2 D^Y_{ij} q_{ij} \left( \frac{(1 - \beta) q_{ij}}{\beta p'_{ij} + (1 - \beta) q_{ij}} + \log \left( \beta p'_{ij} + (1 - \beta) q_{ij} \right) \right)$$

$$+ \sum_{k \neq l} (1 + (D^Y_{kl})^2)^{-1} 2 D^Y_{ij} q_{ij}^2 \left( \frac{(1 - \beta) q_{kl}}{\beta p'_{kl} + (1 - \beta) q_{kl}} + \log \left( \beta p'_{kl} + (1 - \beta) q_{kl} \right) \right)$$

$$= 2 D^Y_{ij} \left( -(1 + (D^Y_{ij})^2)^{-1} q_{ij} \left( \frac{(1 - \beta) q_{ij}}{\beta p'_{ij} + (1 - \beta) q_{ij}} + \log \left( \beta p'_{ij} + (1 - \beta) q_{ij} \right) \right) \right)$$

$$+ 2 D^Y_{ij} q_{ij}^2 \sum_{k \neq l} (1 + (D^Y_{kl})^2)^{-1} \left( \frac{(1 - \beta) q_{kl}}{\beta p'_{kl} + (1 - \beta) q_{kl}} + \log \left( \beta p'_{kl} + (1 - \beta) q_{kl} \right) \right)$$

$$= 2 D^Y_{ij} q_{ij} (1 + (D^Y_{ij})^2)^{-1} \left( - \frac{(1 - \beta) q_{ij}}{\beta p'_{ij} + (1 - \beta) q_{ij}} - \log \left( \beta p'_{ij} + (1 - \beta) q_{ij} \right) \right.$$

$$\left. + \sum_{k \neq l} q_{kl} \left( \frac{(1 - \beta) q_{kl}}{\beta p'_{kl} + (1 - \beta) q_{kl}} + \log \left( \beta p'_{kl} + (1 - \beta) q_{kl} \right) \right) \right)$$

Finally we combine them into the derivative of $\text{JS}_\beta^\alpha$ with respect to $D^Y_{ij}$

$$\frac{(1 + (D^Y_{ij})^2)}{2 D^Y_{ij} q_{ij}} \frac{\delta \text{JS}_\beta^\alpha (P' \| Q)}{\delta D^Y_{ij}} = -\alpha \left( - \frac{\beta p'_{ij}}{\beta q_{ij} + (1 - \beta) p'_{ij}} + \sum_{k \neq l} \frac{\beta p'_{kl} q_{kl}}{\beta q_{kl} + (1 - \beta) p'_{kl}} \right)$$

$$+ (1 - \alpha) \left( -(1 + \log q_{ij}) + \sum_{k \neq l} q_{kl} (1 + \log q_{kl}) \right)$$

$$- (1 - \alpha) \left( - \frac{(1 - \beta) q_{ij}}{\beta p'_{ij} + (1 - \beta) q_{ij}} - \log \left( \beta p'_{ij} + (1 - \beta) q_{ij} \right) \right)$$

$$- (1 - \alpha) \sum_{k \neq l} q_{kl} \left( \frac{(1 - \beta) q_{kl}}{\beta p'_{kl} + (1 - \beta) q_{kl}} + \log \left( \beta p'_{kl} + (1 - \beta) q_{kl} \right) \right)$$

$$= \frac{\alpha \beta p'_{ij}}{\beta q_{ij} + (1 - \beta) p'_{ij}} - \sum_{k \neq l} \frac{\alpha \beta p'_{kl} q_{kl}}{\beta q_{kl} + (1 - \beta) p'_{kl}}$$

$$+ (1 - \alpha) \left( -(1 + \log q_{ij}) + \frac{(1 - \beta) q_{ij}}{\beta p'_{ij} + (1 - \beta) q_{ij}} + \log \left( \beta p'_{ij} + (1 - \beta) q_{ij} \right) \right)$$

$$+ (1 - \alpha) \sum_{k \neq l} q_{kl} \left( 1 + \log q_{kl} - \frac{(1 - \beta) q_{kl}}{\beta p'_{kl} + (1 - \beta) q_{kl}} - \log \left( \beta p'_{kl} + (1 - \beta) q_{kl} \right) \right)$$

and substitute this result in the gradient with respect to $y_i$

$$\frac{\delta \text{JS}_\beta^\alpha(P' \mid\mid Q)}{\delta y_i} = 2\sum_j \frac{\delta \text{JS}_\beta^\alpha(P' \mid\mid Q)}{\delta D_{ij}^Y} \frac{(y_i - y_j)}{D_{ij}^Y}$$

$$= 4\sum_{j \neq i}(y_i - y_j)(1 + (D_{ij}^Y)^2)^{-1}q_{ij}\left(\frac{\alpha\beta p'_{ij}}{\beta q_{ij} + (1-\beta)p'_{ij}} - \sum_{k \neq l}\frac{\alpha\beta p'_{kl}q_{kl}}{\beta q_{kl} + (1-\beta)p'_{kl}}\right.$$

$$+ (1-\alpha)\left(-(1 + \log q_{ij}) + \frac{(1-\beta)q_{ij}}{\beta p'_{ij} + (1-\beta)q_{ij}} + \log\left(\beta p'_{ij} + (1-\beta)q_{ij}\right)\right)$$

$$\left. + (1-\alpha)\sum_{k \neq l}q_{kl}\left(1 + \log q_{kl} - \frac{(1-\beta)q_{kl}}{\beta p'_{kl} + (1-\beta)q_{kl}} - \log\left(\beta p'_{kl} + (1-\beta)q_{kl}\right)\right)\right).$$

### A.1.2 BOUNDING pJSD

Here, we will provide the proof for the upper bound of the pJSD following the idea of Yamano (2019).

**Theorem** (Upper bound on parametrized JS divergence) *For $0 \leq \alpha \leq 1$ and $0 < \beta < 1$ the parametrized JS divergence is bounded from above by*

$$\text{JS}_\beta^\alpha \leq -\log(1 - \beta).$$

*Proof.* Let us first define a known bound for general f-divergences Yamano (2019).

**Lemma 4.** *Let $f^*(x)$ be the conjugate function of $f(x)$ defined as $f^*(x) = xf\left(\frac{1}{x}\right)$. The f-divergence satisfies the upper bound*

$$D_f(P \mid\mid Q) \leq \lim_{x \to 0}\left(f(x) + f^*(x)\right).$$

We will use the definition of our parametrized Jensen Shannon Divergence

$$\text{JS}_\beta^\alpha = \alpha\text{KL}(P \mid\mid \beta Q + (1-\beta)P) + (1-\alpha)\text{KL}(Q \mid\mid P + (1-\beta)Q)$$

to bound the individual terms, which are essentially skewed KL divergences, to then obtain the upper bound for pJSD. The skewed $\beta$-KL divergence is given by

$$\text{KL}_\beta(P \mid\mid Q) = \sum_{x \in X} p(x) \log \frac{p(x)}{(1-\beta)p(x) + \beta q(x)}.$$

Choosing $f_1(x) = x\log\frac{x}{(1-\beta)x+\beta}$, we can see that the skewed $\beta$-KL divergence belongs to the family of f-divergences. Similarly, choosing $f_2(x) = -\beta\log\left(\beta x + 1 - \beta\right)$, we get the reverse $\beta$-KL divergence $\text{KL}_\beta(Q \mid\mid P)$. We thus get

$$\text{JS}_\beta^\alpha = \alpha\text{KL}_\beta(P \mid\mid Q) + (1-\alpha)\text{KL}_\beta(Q \mid\mid P)$$

$$\leq \alpha \lim_{x \to 0}\left(f_1(x) + f_1^*(x)\right) + (1-\alpha)\lim_{x \to 0}\left(f_2(x) + f_2^*(x)\right)$$

$$= \alpha \lim_{x \to 0}\left(x\log\frac{x}{(1-\beta)x + \beta} + \log\frac{1}{1 - \beta + x\beta}\right)$$

$$+ (1-\alpha)\lim_{x \to 0}\left(-\log\left(\beta x + 1 - \beta\right) - x\log\left(\frac{\beta}{x} + 1 - \beta\right)\right)$$

$$= \alpha\log\frac{1}{1 - \beta} + (1-\alpha) - \log(1 - \beta)$$

$$= -\log(1 - \beta),$$

where the first inequality is achieved by bounding the individual skewed $\beta$-KL divergences using Lemma 4, the following equality is obtained by plugging in the definitions of the corresponding f-divergences $f_1$ and $f_2$ with their respective conjugate, and the remainder is obtained by plugging in $x = 0$ and using basic math. This completes the proof. $\qquad\square$

### A.1.3 CONFETTI MAINTAINS METRIC PROPERTIES

**Theorem 5.** *Assuming that $D^X$ and $D^Z$ are based on valid metrics, for any $\lambda > 0$, $(D^X \ominus_\lambda D^Z)$ fulfills the metric axioms of non-negativity, symmetry, identity, and the triangle inequality.*

*Proof.* We will prove that

$$f_\lambda(i,j) = \begin{cases} -\frac{1}{2}\lambda D_{ij}^Z + \lambda & i \neq j, \\ 0 & i = j, \end{cases}$$

fulfills the metric axioms and use the fact that the sum of two metrics is a metric.

1. Non-negativity: $\forall i, j. f_\lambda(i,j) \geq 0$.
   Using the definition of $f_\lambda(i,j)$, we get $\lambda \geq \frac{1}{2}\lambda D_{ij}^Z$. Since $D_{ij}^Z \in (0,1)$, we know that the inequality always holds.

2. Symmetry: $\forall i, j. f_\lambda(i,j) = f_\lambda(j,i)$.
   Follows directly from the symmetry property of $D^Z$.

3. Identity: $\forall i, j. f_\lambda(i,j) = 0 \iff i = j$.
   We know that for $i \neq j$, $f_\lambda(i,j) \in [-\frac{1}{2}\lambda D_{max}^Z + \lambda, -\frac{1}{2}\lambda D_{min}^Z + \lambda]$. Using $D_{ij}^Z \in (0,1)$, we get $f_\lambda(i,j) \in [\frac{1}{2}\lambda, \lambda]$. The other direction follows directly from the definition.

4. Triangle Inequality: $\forall i, j, k. f_\lambda(i,j) \leq f_\lambda(i,k) + f_\lambda(k,j)$.
   Using the insight $f_\lambda(i,j) \in [\frac{1}{2}\lambda, \lambda]$ from before, we get $f_\lambda(i,j) \leq \max_{x,y}(f_\lambda(x,y)) = \lambda = 2 \cdot \frac{1}{2}\lambda = 2 \cdot \min_{x,y}(f_\lambda(x,y)) \leq f_\lambda(i,k) + f_\lambda(k,j)$.

Thus, $(D^X \ominus_\lambda D^Z)$ implements a metric. $\qquad\square$

### A.1.4 CONFETTI IS ROBUST AGAINST UNINFORMATIVE PRIORS

**Theorem 6.** *Assume the prior is uninformative, that is $D^Z \perp\!\!\!\perp D^X$. Furthermore, the distances are normalized to $D_{ij}^Z \in [0,1]$, which is ensured by our method. For fixed $\lambda > 0$ let $(F_\lambda)_{ij} = D^X \ominus_\lambda D^Z$ be the high dimensional distances with factored out prior. On expectation, the neighbourhoods of $X$ and $X$ with factored out prior are the same, that is $N_k^{F_\lambda}(i) =_{E[.]} N_k^{D^X}(i)$, for any $i$ and $k$.*

*Proof.* Let us first look at what on expectation the distance of some nearest point of $i$ is under $F_\lambda$. We now pick the $j \in N_k^{F_\lambda}(i)$ with the largest distance. Note that the distances $D$ define a fixed ordering and thus for each $j$ there is a $k'$ such that $j$ has the largest distance in $N_{k'}^{F_\lambda}(i)$. We prove by contradiction. Let $j \notin N_k^{D^X}(i)$, hence there are $k$ indices given by $L = N_k^{D^X}(i)$ with $\forall l \in L. D_{il}^X < D_{ij}^X$. Let $P_X$ and $P_Z$ be the generating distributions of $D_X$ and $D_Z$, respectively. We get

$$D_{il}^X < D_{ij}^X$$

$$D_{il}^X - \frac{1}{2}\lambda E_{P_Z}[D_{il}^Z] + \lambda < D_{ij}^X - \frac{1}{2}\lambda E_{P_Z}[D_{il}^Z] + \lambda$$

$$E_{P_Z}[D_{il}^X - \frac{1}{2}\lambda D_{il}^Z + \lambda] < E_{P_Z}[D_{ij}^X - \frac{1}{2}\lambda D_{ij}^Z + \lambda]$$

$$E_{P_Z}[(F_\lambda)_{il}] < E_{P_Z}[(F_\lambda)_{ij}],$$

where line 1 to 2 is obtained using that $D^X \perp\!\!\!\perp D^Z$ and $E_{P_Z}[D_{il}^Z] \in [0,1]$, and line 2 to 3 is obtained by using linearity of expectation and $D^X \perp\!\!\!\perp D^Z$. In other words, for all $l$ on expectation the adapted distance to $i$, $(F_\lambda)_{il}$, is smaller than the adapted distance between $i$ and $j$, $(F_\lambda)_{ij}$. This means that $L$ contains $k$ indices that have smaller distances under $F_\lambda$ than $j$, hence $j \notin N_k^{F_\lambda}(i)$. This is a contradiction. $\qquad\square$

A.1.5    CONFETTI PSEUDOCODE

---

**Algorithm 1:** CONFETTI

---

**Data:** Data distances $D^X$, prior distances $D^Z$, penalty $\lambda$
**Result:** Adjusted distances $D^X \ominus_\lambda D^Z$

1  $D^X \leftarrow \frac{D^X}{\max D^X}$
2  $D^Z \leftarrow \frac{D^Z}{\max D^Z}$
3  **foreach** $i \in [1, n]$ **do**
4      **foreach** $j \in [i + 1, n]$ **do**
5          $(D^X \ominus_\lambda D^Z)_{ij} \leftarrow D^X_{ij} - \frac{1}{2}\lambda D^Z_{ij} + \lambda$
6          $(D^X \ominus_\lambda D^Z)_{ji} \leftarrow (D^X \ominus_\lambda D^Z)_{ij}$

7  **return** $(D^X \ominus_\lambda D^Z)$

---

# B    EXPERIMENT APPENDIX

## B.1    INVERSE SUPERVISED LLE (sLLE$^{-1}$)

In SLLE (De Ridder et al., 2003), class labels given as additional input are used to emphasize the structure within the classes. Thus, SLLE improves the separation between classes by homogenizing the local neighbourhood of each sample with respect to the class labels. This is done by pushing away data points with different class labels, increasing the distance of such points by a fraction of the maximum distance observed in the data. This can be summarized by the following update procedure for the sample distances, using the same notation as in the main paper

$$D' = D^X + \alpha \max(D^X)\hat{D}, \ \ 0 \leq \alpha \leq 1,$$

with $\hat{D}$ the binary matrix with $\hat{D}_{ij} = 1$ if samples $x_i$ and $x_j$ are from different classes.

In our work, we are interested in factoring out the prior, which is essentially the opposite of SLLE. By manipulating the objective from above, we can however achieve our goal of factoring our prior knowledge, by optimizing for

$$D' = D^X + \alpha \max(D^X)(1 - \hat{D}), \ \ 0 \leq \alpha \leq 1.$$

We call this method sLLE$^{-1}$. This method can still only deal with labels as input, but for that it works surprisingly well in our experiments, yielding better results than the current state of the art ctSNE.

## B.2    CHOOSING THE HYPERPARAMETERS

All methods involved in the experiment section come with a set of hyperparameters that the user has to fix. We decided to generate a small synthetic data set, generated differently than the one used for our experiments, to settle for a good set of parameters for each of the methods. In particular, we explore for CONFETTI coupled with tSNE the parameter $\lambda \in \{0.2, 0.4, 0.5, 0.6, 0.8, 1, 1.2, 1.4, 1.5, 1.6, 1.8, 2, 2.5, 3, 4\}$ and perplexity $30, 50, 200, 400$, for JEDI we search for $\alpha \in \{0, 0.2, 0.5, 0.8, 1\}$, $\beta \in \{0.8, 0.99, 1\}$, and perplexity and prior perplexity $50, 200, 400$ each. For ctSNE we search for $\beta \in \{1e^{-2}, 1e^{-3}, ..., 1e^{-7}\}$, and for sLLE$^{-1}$ the number of neighbours $k \in \{10, 30, 50, 100, 200, 400, 500, 600\}$ and parameter $\alpha \in \{0.0001, 0.001, 0.1, 0.2, ..., 1\}$. We note that the authors of ctSNE suggest to use a small value $\beta = 0.01$, but use $\beta = 0.0001$ for their experiment, which led us to the search grid for their method.

We generated a simple 10-dimensional dataset with 1k data points, with samples assigned to one of four clusters in the first 4 dimensions, and one of two clusters in dimension 5 and 6. The clusters in the first 4 dimensions are centered at the four unit vectors along the four axes of the dimensions. The clusters in the other dimensions are centered at $\left(\begin{smallmatrix}\frac{1}{3}\\0\end{smallmatrix}\right), \left(\begin{smallmatrix}0\\\frac{1}{3}\end{smallmatrix}\right)$. The remaining four dimensions are gaussian noise $\mathcal{N}(0, 1)$. Each data point is randomly assigned to one cluster of the first four, and

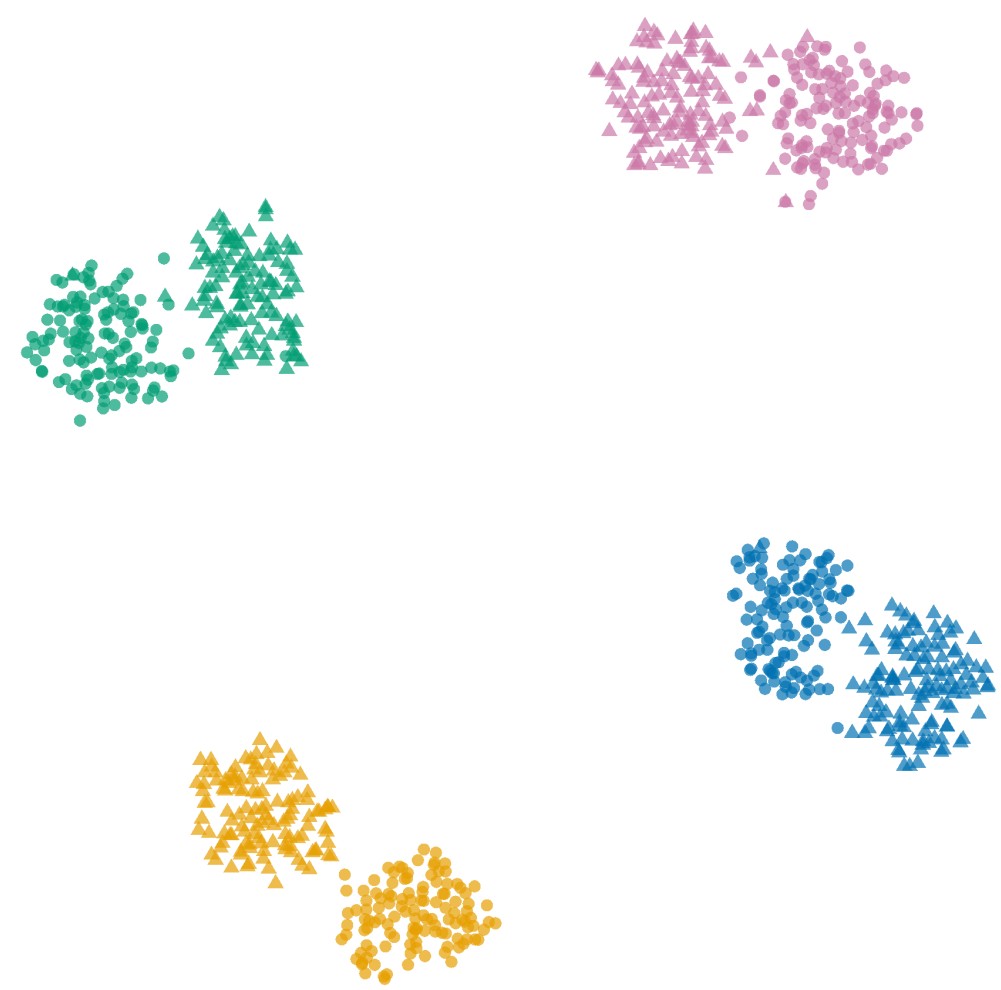

Figure 4: Visualization of the synthetic data set for parameter tuning based on tSNE. We see samples with the same colors clustered together (clustering planted in dimensions A) as well as samples with the same shape (clustering planted in dimensions B).

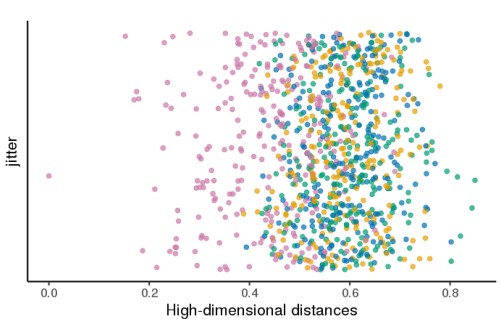

(a) Euclidean distances to all other samples computed over all dimensions.

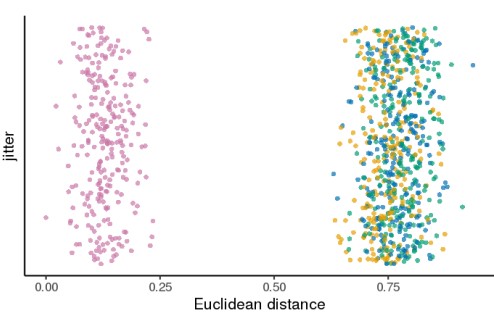

(b) Euclidean distances to all other samples computed only over dimensions d1 to d4. We observe the clustering planted into those dimensions.

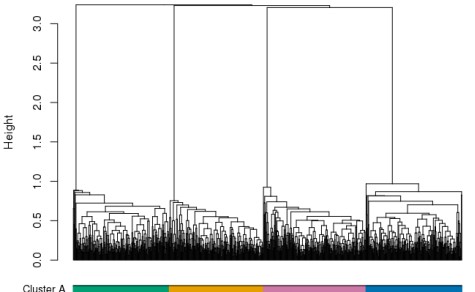

(c) Hierarchical clustering with average linkage based on dimensions d1 to d4. The four different clusters are easily identifiable.

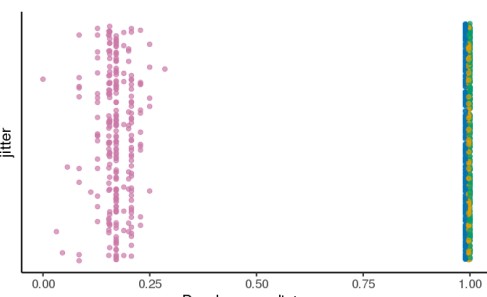

(d) Scaled dendrogram distances from the pink reference sample to all other samples. It is defined as the height of their lowest common ancestor node in the dendrogram given in (c).

Figure 5: Visualization of different distances for one sample from the pink cluster to all other samples in the synthetic data set for parameter tuning.

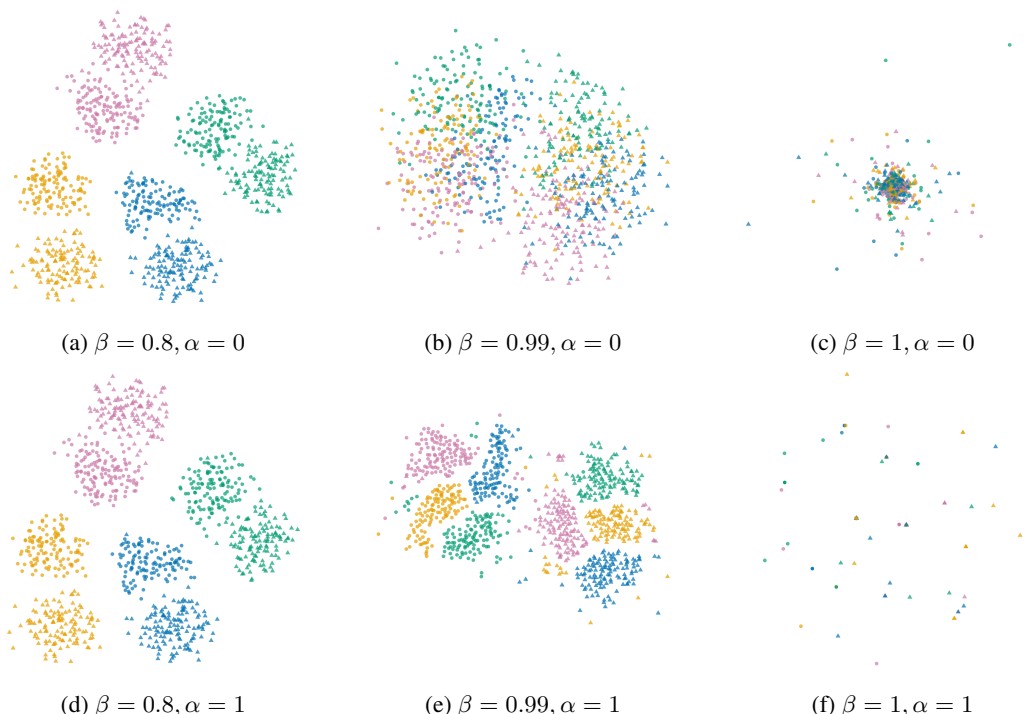

Figure 6: Visualizations of the synthetic data set for parameter tuning by JEDI with a Euclidean distance prior based on the dimensions in A (color clusters). Here we compare the influence of the smoothing parameters $\alpha$ and $\beta$.

one cluster in dimension 5 and 6, and takes the coordinates of the cluster centers plus some gaussian noise $\mathcal{N}(0, 0.01)$. A tSNE visualization is given in App. Fig. 4. For each method, we provide embeddings for a subset of their search grid that gives a good idea how the embedding changes when varying the parameters in Figures 5, 6, 7, 8, 9, 10, and 11.

Looking at the results, we observe that for CONFETTI, as expected, the gradual increase in $\lambda$ is directly reflected in the embedding with how strongly differently colored points merge. Here, we settle for a perplexity of 50 for the prior and $\lambda = 2$, as this allows to take into account much information of the prior while yielding nicely separated and mixed colored clusters.

For JEDI, we have to investigate a good setting of 4 different parameters in a principled manner. Fixing the perplexities, we observe that $\alpha = 0$ and $\beta = 0.99$ yield much better mixtures of the similarly colored points, than any other combination, with JEDI not converging with $\beta = 1$ (App. Fig. 6). Yamano (2019) already pointed out that $\beta = 0.99$ worked well as a choice for the divergence mixing. Fixing $\beta = 0.99$ we proceed to analyse the impact of $\alpha$ and the perplexity parameters, noting that the latter is highly dependent on the data set size, and should be adjusted by the user as with original tSNE. Here, we settle for $\alpha = 0$, perplexity of $\frac{1}{5}n$ and prior perplexity $\frac{1}{10}n$, where $n$ is the number of samples.

For ctSNE, $\beta = 1e^{-5}$ worked best, which is also close to what the authors use for their original experiments. In the case of sLLE$^{-1}$, we observe that for $\alpha \geq 0.3$, independent of $k$, similarly colored points are not clustered together anymore. Furthermore, only with $k \geq 200$ the two desired clusters are clearly visible. We thus propose to use $\alpha = 0.5$ and $k = \frac{1}{2}n$ as default parameters, where $k$ should be carefully adjusted depending on the data set at hand.

## B.3 SYNTHETIC DATA

The data has 14 dimensions in total, where each sample belongs to one of 4 clusters (A1-A4) in dimension 1-8 and one of four clusters in dimension 9-12 (B1-B4). We first draw cluster centers

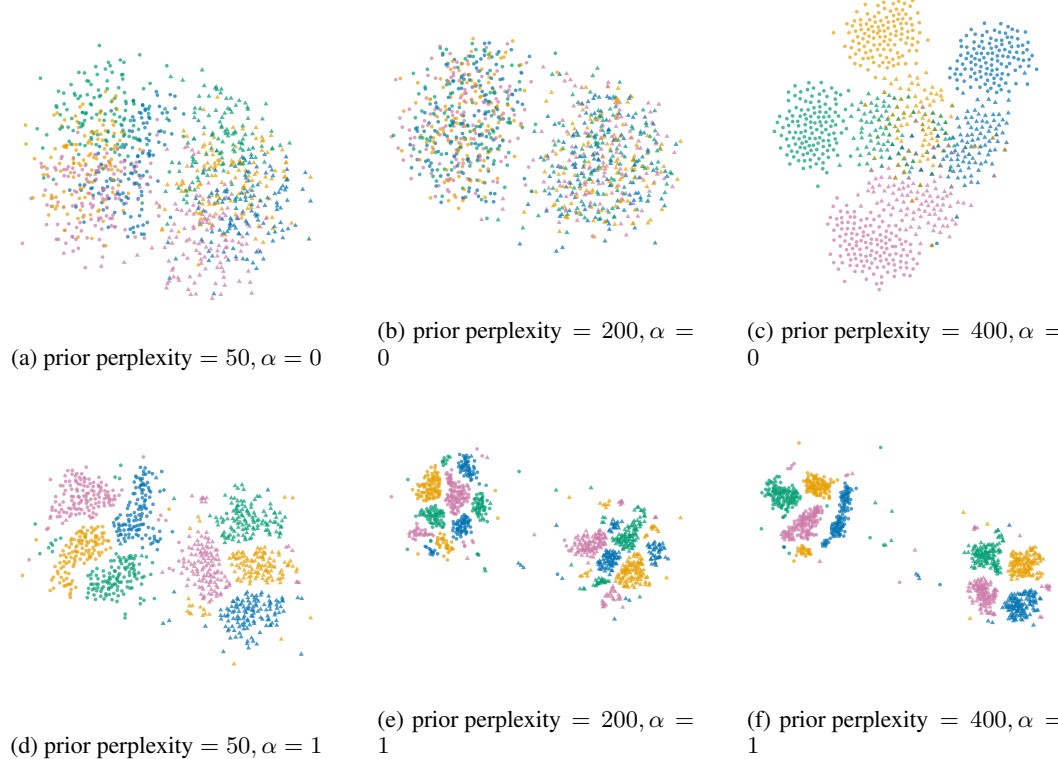

(a) prior perplexity $= 50, \alpha = 0$

(b) prior perplexity $= 200, \alpha = 0$

(c) prior perplexity $= 400, \alpha = 0$

(d) prior perplexity $= 50, \alpha = 1$

(e) prior perplexity $= 200, \alpha = 1$

(f) prior perplexity $= 400, \alpha = 1$

Figure 7: Visualizations of the synthetic data set for parameter tuning by JEDI with a Euclidean distance prior based on the dimensions in A (color clusters). We compare the effect of an increasing prior perplexity on both divergences determined by $\alpha$.

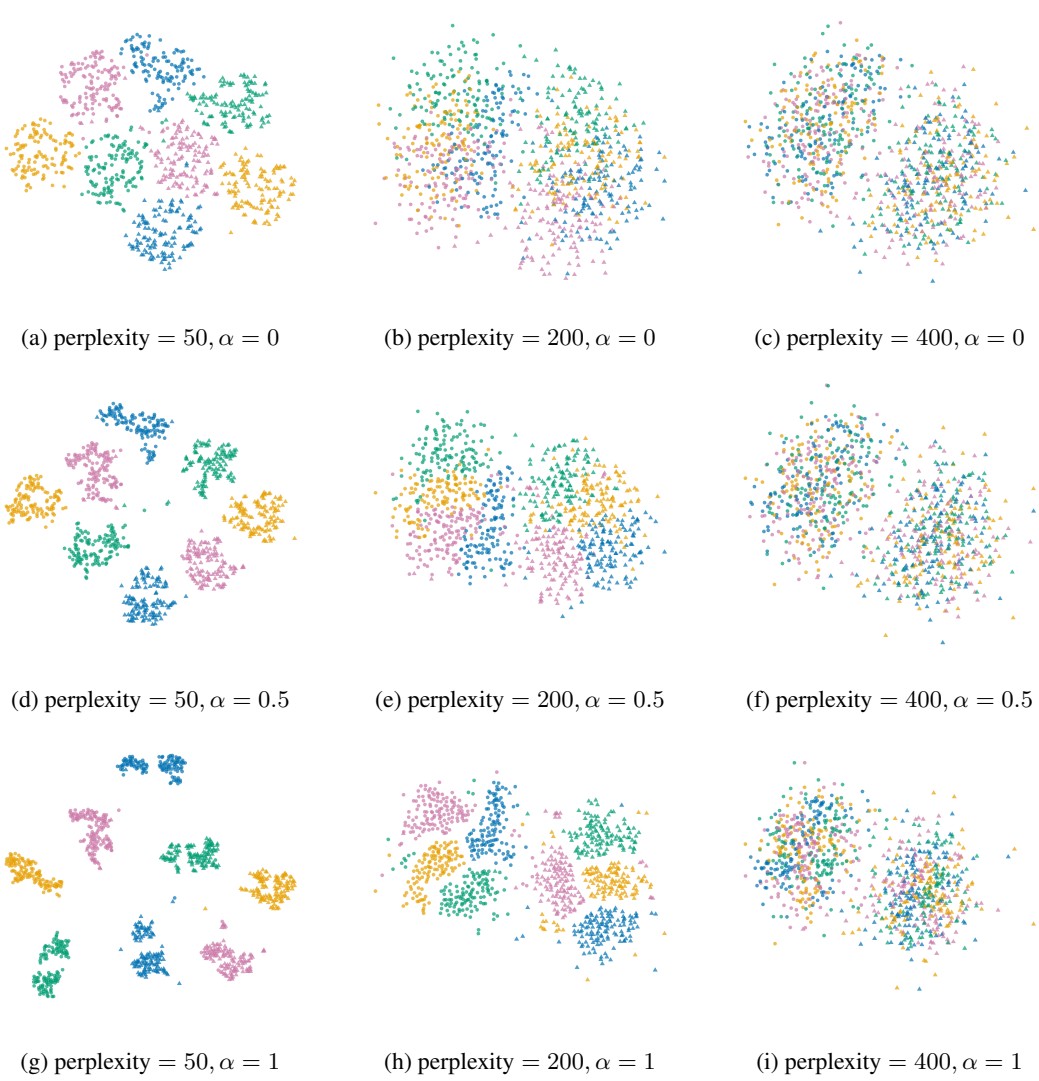

Figure 8: Visualizations of the synthetic data set for parameter tuning by JEDI with a Euclidean distance prior based on the dimensions in A (color clusters). We see the effect of different perplexities in combination with different values for the mixture of divergences $\alpha$.

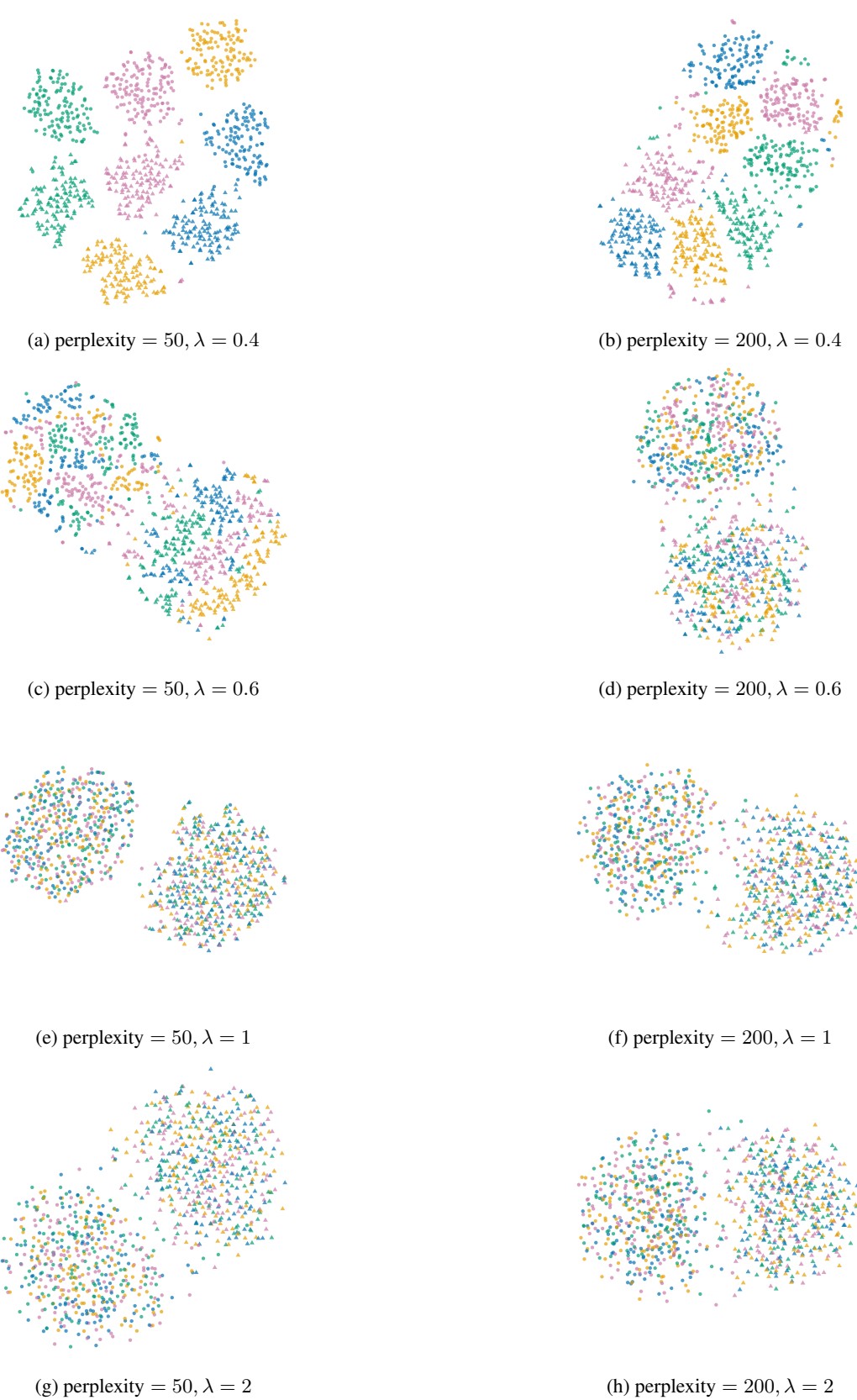

Figure 9: Visualizations of the synthetic data set for parameter tuning by CONFETTI with a Euclidean distance prior based on the dimensions in A (color clusters). For two different perplexity values we increase the maximum penalty $\lambda$ from 0.4 to 2.

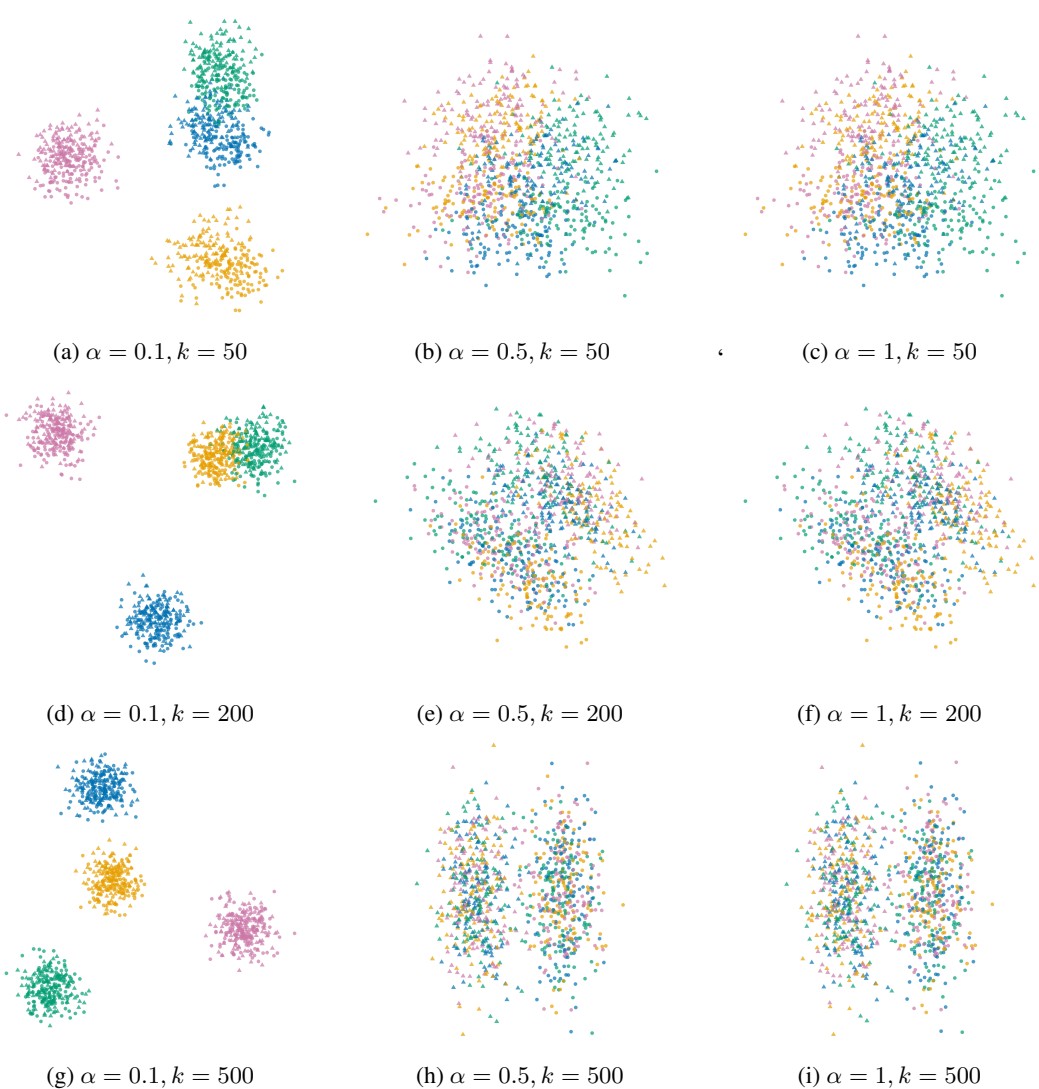

Figure 10: Visualizations by sLLE$^{-1}$ with labels prior based on the clusters in A (color). The amount how much the distances are adjusted is determined by $\alpha$ and the number of neighbors to consider is $k$.

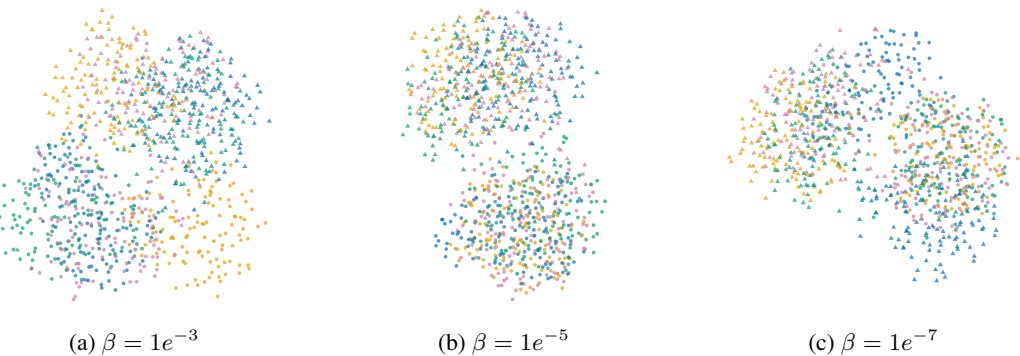

Figure 11: Visualizations by ctSNE with a label prior based on the clusters in A (color). We try different values for $\beta$ which influences how strong the prior is factored out.

from $\mathcal{N}(0, 2)$ for each of the eight clusters. Then, the feature values for each sample are generated in three steps as follows:

1. Pick a cluster a from A1-A4 with probability 0.1, 0.2, 0.3, or 0.4, respectively. Add Gaussian noise to the cluster center a with standard deviation 0.1, 0.2, 0.3, or 0.4, respectively. Noise is drawn and added for each dimension independently.

2. Pick a cluster a from B1-B4 with probability 0.1, 0.2, 0.3, or 0.4, respectively. Add Gaussian noise to the cluster center a with standard deviation 0.1, 0.2, 0.3, or 0.4, respectively. Noise is drawn and added for each dimension independently.

3. The remaining 2 dimensions of every sample are Gaussian noise from $\mathcal{N}(0, 1)$.

A tSNE visualization of the data is given in App. Fig. 12. For this dataset, we take the euclidean distances between all samples in the first 8 dimensions as prior for our methods, and the ground truth label assignment A1-A4 as prior for the other methods. Here, we additionally provide the embeddings of CONFETTI and JEDI with label assignment prior, given in App. Fig. 13, and the NOS plots for both the input data and the prior in App. Fig. 14. While the NOS plots and corresponding area between curves give insight in how well an embedding factors out or maintains certain information, it is hard to optimize. In particular, it unclear how to combine and balance the two scores for how much prior is removed and how much information of the input is maintained. Even if these issues are resolved, the main problem will be that the kNNs that define this score are essentially describing binary relationships between entities, thus optimizing for optimal neighbourhoods does not give any clue how to arrange the samples in a 2D continuous space.

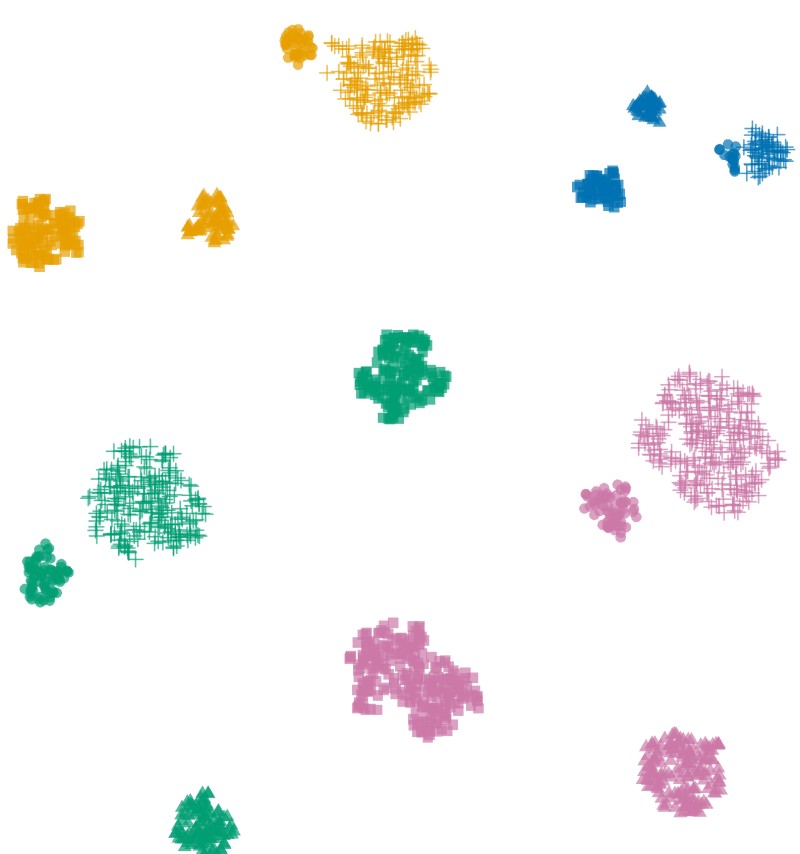

Figure 12: *t*SNE *of synthetic data.* Visualized is a tSNE plot for the synthetic data set using perplexity 50. Color corresponds to cluster assignment in dimension 1-8 and shape corresponds to cluster assignment in dimension 9-12.

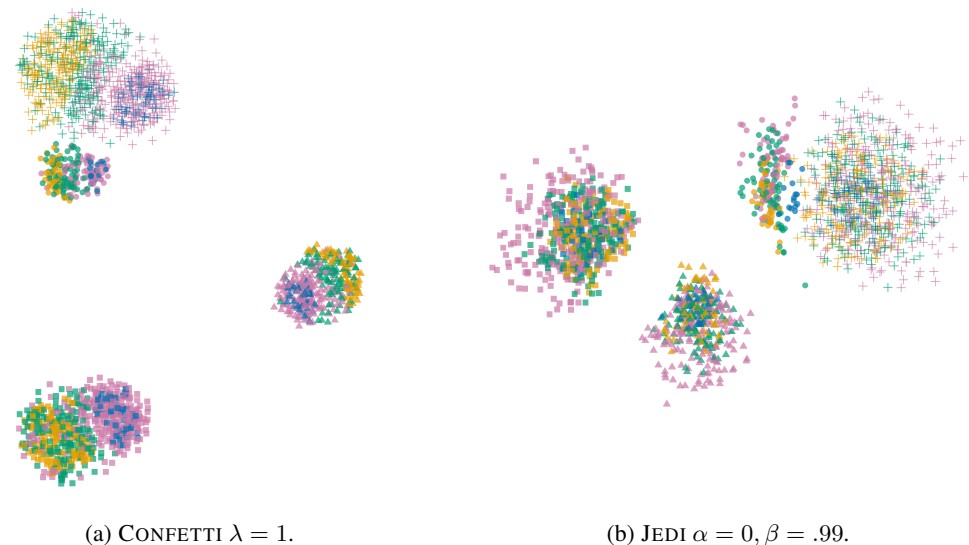

(a) CONFETTI $\lambda = 1$.  (b) JEDI $\alpha = 0, \beta = .99$.

Figure 13: JEDI *and* CONFETTI *with label prior.* Visualized are the plots generated by our methods when fed with the ground truth labels corresponding to the cluster assignment in the first 8 dimensions of the synthetic data set. Color corresponds to cluster assignment in dimension 1-8 and shape corresponds to cluster assignment in dimension 9-12.

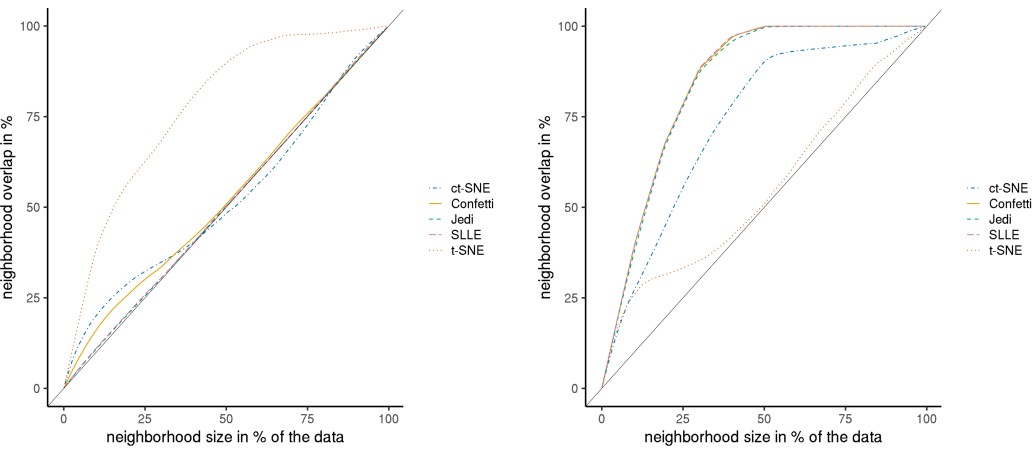

(a) NOS plot between prior labels and embedding (smaller is better).  (b) NOS plot between non-prior dimensions and embedding (larger is better).

Figure 14: *NOS score visualizations.* Given are the NOS plots for prior labels and embedding (a) and non-prior dimensions 9-12 and embedding (b) on the synthetic data set. On the x-axis, the neighbourhood $k$ is varied between 0% and 100% of data points, the y-axis is the NOS score.

## B.4 FLOWER DATA

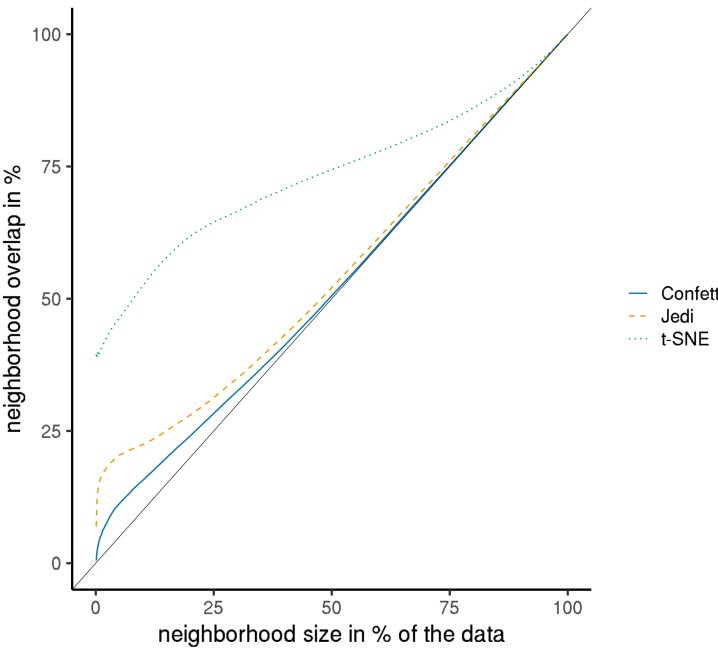

Figure 15: *NOS plot for flowers.* Visualized is the NOS score for the Oxford flower data set for JEDI and CONFETTI against the prior. On the x-axis, the neighbourhood $k$ is varied between 0% and 100% of data points, the y-axis is the NOS score.

## B.5 SINGLE CELL DATA

Here, we provide additional plots for the single cell experiments. In particular, we carried out an agglomerative clustering based on marker gene expression, which arrives at the same number of clusters as the original paper (App. Fig. 16b) Additionally, we show how the ctSNE embedding looks when coloring the visualization according to cell type of the original paper (App. Fig. 16a), and a new result when we use tissue information as prior knowledge for CONFETTI (App. Fig. 17a), as well as the vanilla UMAP plot with coloring according to batch ID.

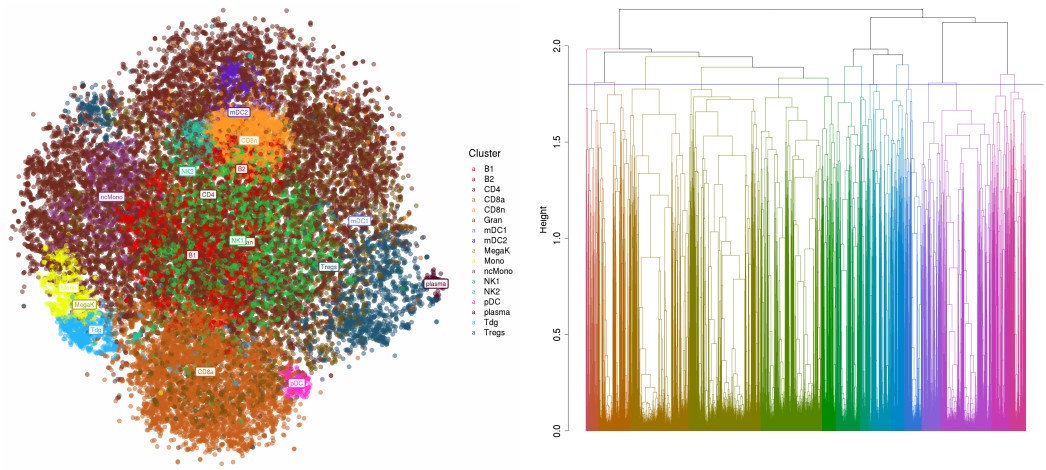

(a) ctSNE embedding on single cell data with color accordint to cell labels.

(b) Dendogram for complete linkage clustering of samples according to marker gene expression.

Figure 16: *Embedding of ct*SNE *and clustering according to marker genes.* a) Visualized is the embedding obtained from ctSNE for the single cell data with our cluster labels as prior. The coloring is according to cell type. b) Dendogram of agglomerative clustering with complete linkage on marker gene expression. A natural cutting point is at 1.8 (blue horizontal line), which retrieves the same number of clusters as the orginal paper.

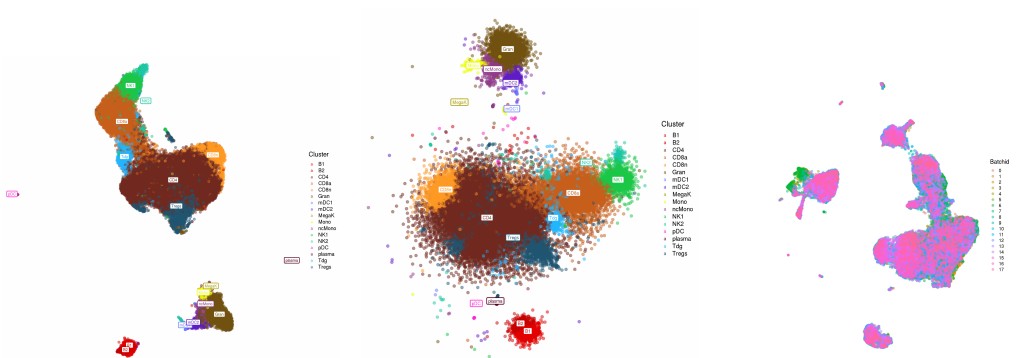

(a) Embedding of CONFETTI with tissue prior.

(b) Embedding of JEDI with marker gene prior.

(c) Embedding of vanilla UMAP with batch ID coloring.

Figure 17: *Additional SC embeddings.* Visualized is the embedding of CONFETTI for the single cell data with the tissue type (blood vs CSF) as prior with coloring according to cell labels (a), the embedding obtained from JEDI for the single cell data with marker gene expression as prior, with coloring is according to the cell labels (b), and the original UMAP embedding with coloring according to batch ID (c).

