# OpenReview forum: "Factoring out Prior Knowledge from Low-Dimensional Embeddings"
_ICLR.cc/2021/Conference — Reject_

### Official Review · AnonReviewer4 · 2020-10-23

**Rating:** 5
**Confidence:** 3

**Review:**

Summary:

The author(s) provide two methods for factoring out specific covariates from tSNE, UMAP or other distance matrices. The first one is JEDI, an extension of tSNE that minimizes a parameterized divergence (that takes into account the information to be factored out) instead of the simple KL divergence between high dimensional data and low-dimensional embedding. Because tSNE has inherent limitations, the author(s) also propose CONFETTI, a simple approach to create a distance matrix based on the distance matrices of the covariates and the input data. This produces a proper distance metric and can be used upstream of any embedding procedure. On synthetic data, the proposed methods perform to the level of ctSNE as well as a second baseline sLLE-1. However, both methods are not directly meant to solve the original problem. Only JEDI and CONFETTI may factor out continuous variations (ctSNE is designed for discrete clusters). On real world data, the method seems to effectively reorganize embeddings (either of images, or of single cells) by factoring out variations of interest.

Strong points:
1. The paper is well-written and simple to understand
2. The idea is conceptually interesting

Weak points:
1. The manuscript does not acknowledge foundational work. The idea of factoring nuisance variation out of representations is rather common, either in the setting of matrix factorization, Gaussian processes or variational autoencoders [1, 2]. This is especially true in the setting of high-dimensional genomics data, in which all the work around "removing unwanted variations"  (RUV) became a common theme (> 10 papers).
2. I am skeptical of the applicability of the method, especially regarding runtime. Much of the paper describes JEDI. However, JEDI takes 100 hours to run on the scRNA-seq data compared to tSNE (that should finish in less than 10 minutes, or at least be faster than ctSNE that runs in 20 minutes).
3. The construction of the distance matrix for CONFETTI is interesting, but there is no effort in understanding in which sense such construction is efficient, optimal or even why it seems to be a reasonable choice. I do not think that the presented theory is helpful for understanding when the method would work well or not.
4. In terms of experiments, a stronger point could be made by choosing a challenging setting (removing cell-cycles in single-cell RNA-sequencing data in order to better recover cell types as in [2]), and comparing against RUV models, to justify why corrections on the distance matrices is more efficient than linear corrections on the input space

[1] Gagnon-Bartsch, J., Jacob, L. & Speed, T.P. Removing unwanted variation from high dimensional data with negative controls. Tech. Rep. 820, Department of Statistics, University of California, Berkeley (2013).
[2] Buettner, F. et al. Computational analysis of cell-to-cell heterogeneity in single-cell RNA-sequencing data reveals hidden subpopulations of cells. Nat. Biotechnol. 33, 155–160 (2015).

---

> ### Author Response · Authors · 2020-11-17
> **Answer to concerns of Reviewer #4**
>
> Thank you for your constructive critique.
>
> 1. While this work is related, it is very different in nature. First of all, we want to emphasize that we are interested in developing a general method that is independent of the type of data and does not make assumptions on its distribution.
> Regarding your specific points, we are focusing on obtaining low-dimensional embeddings from high dimensional data. So far, autoencoders have not proven successful in this setting, and neither did Bayesian approaches that assume a distribution. The goal is really to capture the manifold the data lies on in high dimensions, thus take the data as is instead of enforcing a distribution.
> Your reference [1] is indeed very interesting, and we will add it to the related work, yet it is quite distant to what we do. Their approach aims to RUV assuming the input is a linear function of factors of variations, assuming that all of these are known separately, or making assumptions on the distribution of the variables. Furthermore, only factors for which known negative control genes exist can be incorporated. While this might model some tasks related to gene expression well, and thus has its application, it is very different to what we aim to do.
> The second reference [2] is domain specific to gene expression and makes strong assumptions by modelling the gene expression data as a normal distribution.
> As mentioned, here we want to keep the original distribution of the input data intact not posing any assumptions on the shape of the input, and want to be domain independent. While there exist striking differences to our approach, we value your feedback and see the relation to what we do. Thus, we incorporate an additional paragraph addressing RUV, its applications, and the relation to our work in the upcoming draft.
>
> 2. The experiments show that JEDI is applicable in many settings, but similar to the original tSNE it does have problems to scale to very large data. It is important to note, however, that we currently only have a simple, unoptimized implementation of JEDI: our focus here is on proposing a novel approach that lends itself for natural interpretation and is based on information theoretic grounds. As there exist advanced algorithmic solutions to speed up tSNE (e.g. FItSNE) that do permit analysis of single cell data, it is likely that it is possible to also speed up JEDI, but this is beyond the scope of this paper – especially as with Confetti we already have a solution that scales to the single cell setting.
>
> 3. Since there is no clear objective to optimize, as it would be in a supervised setting, this is our suggestion to tackle this problem. There might indeed exist other formulations of the overall problem (such as in the line of RUV), but we are not aware of any general formulation that does keep the manifold of the original data intact, which is crucial for the low-dimensional embeddings.
> This particular formulation of Confetti, has several advantages over e.g. more complex non-linear formulations (as also stated in response to 4. of Reviewer 1). For the downstream embedding process, it is crucial that the input distance matrix does reflect a proper distance metric, as we optimize the low-dimensional distance between each pair of samples, thus we need to have both symmetry and triangle equality in order to successfully optimize.
> Furthermore, we achieve an easy to interpret correspondence between input and prior distance and hence the effect of prior on input. Furthermore, it is crucial to be robust against uninformative priors, such that no knowledge is lost and no false knowledge is generated when looking at the resulting embedding. Both the metric property as well as the theorem about uninformative priors simultaneously are hard to achieve for non-linear formulations that we considered.
>
> 4. We are not aware of a general RUV approach that does account for prior information (or variance induced by certain variables) in a comparable setting without negative controls and assumptions on relationship between input and prior and distribution of certain variables. If we missed on an RUV method that is as general, and not limited to particular gene expression experiments with known negative controls, please point us to the corresponding work.

---

### Official Review · AnonReviewer1 · 2020-10-28
**Interesting paper + Important problem but the method formulation is somewhat contrived**

**Rating:** 6
**Confidence:** 4

**Review:**

Summary:
=======
The distance metric learned by low-dimensional embeddings typically captures the knowledge that we already know. This paper proposes a principled way of factoring out prior knowledge (in the form of distance matrices) from tSNE and UMAP embeddings. Two algorithms are proposed for factoring out prior knowledge. JEDI (for tSNE embeds) uses a parameterized JS divergence-- the objective is to learn a low-dimensional distance metric that preserves high-dimensional distances but is orthogonal to the prior distance matrix. CONFETTI is the second algorithm which also optimizes a similar objective to JEDI but doesn't employ JS divergence and is algorithm independent, so one can use it for tSNE or UMAP. Results are shown on synthetic and real-world flower and cell-sequencing data and they highlight the superior ability of JEDI and CONFETTI algorithms in factoring-out prior knowledge compared to the baselines.



Comments:
==========
The paper is well written and puts itself nicely in context of previous work. Given the ubiquity of low-dimensional embeddings these days, the paper addresses an important problem of factoring out prior information from the embeddings.


1). The paper doesn't describe the details of optimizing the parameterized JS Divergence (pJSD) metric that they propose. Is it even convex?

2). How is the beta parameter of pJSD chosen?

3). I didn't fully understand why the paper makes a big deal about UMAP, when JEDI is based on tSNE and CONFETTI can work with any embedding formulation? How not just mention a "general embedding."

4). The formulation of the CONFETTI method seems a little arbitrary. Why are the prior distances factored out linearly? Again, no optimization details are provided regarding the CONFETTI method.

5). It seems that, as an application, the proposed methods can also be used for factoring out demographic information from word embeddings. For such applications how would one define the prior distance matrix? The datasets used in the paper and the broader setup seems a little contrived. It would be nice to provide guidelines on how one can readily define such prior distance matrices for other applications.


Typo:

Conclusion: "This shows that both are able applicable to real world..."

---

> ### Author Response · Authors · 2020-11-17
> **Answer to concerns of Reviewer #1**
>
> Thank you for your review, we will address each of your concern in the following.
>
> 1) The objectives of SNE, tSNE and ctSNE, are all not convex, and all are optimized via gradient descent. The same holds for JEDI. We give the gradient of the JEDI objective in the Appendix. We hope that the revised formulation is clearer.
>
> 2) Like all hyperparameters for all considered methods, we optimized $\beta$ using a hold-out dataset, obtaining a value of 0.99.
>
> 3) Indeed, as mentioned in the theory section, Confetti can be combined with any embedding algorithm. For the experiments we had to make a choice for an instantiation. We chose UMAP, as it has replaced tSNE as state of the art in the rapidly growing field of single cell analysis, one of the major areas of application for low dimensional embeddings.
>
> 4) We regard the simplicity of the Confetti formulation as a strength. It is easily interpretable as there is a one to one correspondence between a unit of input distance and unit of prior distance (up to a linear factor). Moreover, the linear transformation allows us to prove the key properties on metricity and uninformative priors. The metric property is crucial when considering multiple samples and moving them relative to each other solemnly based on distances (consider breaking the triangle inequality or symmetry during optimization). As in practice we do not know in how far the prior is represented in the input, the theorem about uninformative priors is also crucial, to guarantee that the resulting embedding is not affected by this kind of prior, and thus no knowledge is lost and no false knowledge is generated.
>
> 5) This is a very interesting suggestion. Let us consider some simple text data that a group of user has generated (e.g. tweets). Then our input $X$ would be the distances between the word embeddings of each pair of tweets, the prior $Z$ would be the distance derived from the demographic property between each pair of users (for instance age difference). Do you have any concrete dataset in mind?
> Generally, we just expect distances between samples as both input and prior, which could be derived by anything. These could be simple Euclidean distances between features that you have as meta information, but also derived from e.g. kernels. Suppose you expect that people with similar name write similarly, and you want to discover trends beyond this known information from the text data embedding. Then your pairwise tweet embedding distances are still $X$, while your prior $Z$ is for example the edit distance between the names of each pair of users.
> We will address that point in the extended discussion section of the revision.

---

> > ### Comment · AnonReviewer1 · 2020-11-20
> > **Thank you.**
> >
> > I have read the authors' rebuttal. They have answered my questions. My evaluation stays the same.

---

### Official Review · AnonReviewer2 · 2020-10-30
**Good ideas but somewhat ad-hoc nature in proposed solution, and limited experiments**

**Rating:** 5
**Confidence:** 4

**Review:**

Description:

This paper aims to "factor out" existing prior knowledge from embeddings, by adapting a tSNE objective or from other methods by operating on input distances. The prior knowledge is assumed to be pairwise distances.

In the tSNE case, a neighborhood distribution is derived from the prior pairwise distances, and a "parameterized Jensen-Shannon divergence" is proposed to measure deviation of the output neighborhood from that. The objective is then a difference of the tSNE objective and that divergence, optimized by gradient descent.

In the general case, it is proposed to edit the pairwise distances of the data by, substracting out the pairwise distance given in the prior information (up to some constants), for all pairs of different points. A metric property of the definition is proven, and a proof that independent prior distances would not on expectation change the neighborhoods.

Experiments compare the result to conditional tSNE and to a modification of supervised LLE, on synthetic gaussian-cluster data and on two real data (flowers and single cell sequencing data).
- On synthetic data, results are better than conditional tSNE but difference to the modified supervised LLE seems small.
- On flower data, there are no comparisons to others except unmodified tSNE, but the proposed methods may be able to show structure beyond the prior as claimed.
- In the single cell sequencing data, the comparisons are to unmofidied UMAP and to ctSNE. The UMAP with modified distances seems to show some batch-d and tissue-type differences.

Evaluation:

The idea of factoring out prior knowledge seems very meaningful, although not completely new.

The approaches here seem reasonable but are somewhat simplistic:
- The tSNE approach seems reasonable but not very striking: essentially it is a weighted sum of two cost functions, and most of the detail is just to ensure the new divergence does not overwhelm the original one.
- The distance editing seems rather ad-hoc; while the result may have a metric property, it is hard to say in what sense this is the "right" way to combine the prior information to the distances.

The experiments are rather limited, which is somewhat disappointing; in the flower case no comparison to ctSNE or SLLE-1 is done, or to SLLE-1 in the single cell sequencing data (too large matrix inversion issues are claimed - could those have been resolved e.g. by computing the result for a subset instead?). The results do seem to show ability to show information beyond the prior knowledge, but for a modern dimensionality reduction paper I would expect a more thorough evaluation.

Comments:

- Neighbor embeddings that explore beyond known annotation have been proposed long before conditional tSNE: e.g. the method of [1] includes extraction of gene expression features unrelated to an ontological annotation, to embed structure not explained by that annotation.

[1] Peltonen J, Aidos H, Gehlenborg N, Brazma A, Kaski S. An information retrieval perspective on visualization of gene expression data with ontological annotation. ICASSP 2010.

- The equation in definition 1 seems to be wrong. In the second divergence term, "P' + (1-beta)Q" is not a proper distribution that sums to 1.

- In the single cell sequencing data, you used marker gene expression for "Z" data, but what was the X data? Clarify.

- Figure 1: marker shapes are far too small to be seen at a normal viewing factor or in a printed paper.

- In experiments authors write that JEDI is "not designed for labels as priors" but in the introduction you write "we consider background knowledge in the form of pairwise distances between samples. This formulation allows
us to cover a plethora of practical instances including labels". In what sense is JEDI not designed for labels as priors?

- The flower experiment seems somewhat biased because the input distance is a linear sum of distances including color-distance: therefore the distance-editing by subtracting the color-distance is of course well matched to this construction.

- In the single-cell sequencing case, to use ctSNE authors "provide it the cluster labels from an agglomerative clustering": why was this not done for the flower data too (and for SLLE-1 too)?

- In the single-cell sequencing case, Figure 3(a) is hard to compare to (b) and (c) since the coloring means different things.

- In Figure 3(c), why are there smooth changes of the batch-id colors? Are batches with close-by bacth numbers somehow expected to be more similar to each other?

---

> ### Author Response · Authors · 2020-11-17
> **Answer to concerns of Reviewer #2**
>
> Thank you for your detailed review.
> First of all, we want to clarify some concerns regarding the experiments. For the synthetic data, we give the competing methods a big advantage by providing them the *ground truth* label information, rather than embedding derived distances provided to our methods. In practice, for most applications (e.g. single cell data) only distances would be available. See also the comment to Reviewer #3.
>
>
> Regarding the other comments:
> - Thank you for the comment on the related work, we were not aware of that specific application. While relevant, it is restricted to a specific type of data and prior, and not directly generalizable to the state of the art embedding approaches. We will add the reference in the upcoming revision.
>
> - Regarding your comment on Equation 1, yes, a $\beta$ was missing. Good catch!
>
> - We will add a clarification on what exactly input $X$ is for the single cell data.
>
> - Regarding marker shape sizes in Figure 1, in the revised draft we will distribute the images across two rows, which should resolve this problem.
>
> - The distance derived from label part is indeed expressed in an unfortunate way. A distance matrix is able to reflect labels and thus can deal with them, but similar to using a binary variable in linear regression, its meaning might be not modeled ideally. We adapted the formulation in the upcoming revision.
>
> - For the flower data, as stated in the paper, the Oxford Flower data was considered as a test for our method to cope with noisy real world distance matrices. We purposely left the color information in that data, so we could be sure that the prior is present in the input data and that the prior is one of the dominating features in the embedding. Since we want to investigate how well a prior is removed, we need to be sure that this prior is present and is given as structure in the embedding.
>
> - The Oxford Flower data only provides distance matrices, so while we can compute labels for the prior similar as for single cell data, we cannot provide SLLE-1 or ctSNE with the *input* data $X$. None of the available implementations for SLLE and ctSNE can deal with distance matrices as input. That is, the issue here is not the prior, but rather the available input data. Especially in settings where only distance matrices are available (e.g. kernel-based distances for unstructured data) existing methods are not applicable.
>
> - Regarding visualizing a subset of the single cell data with SLLE-1, it does not correspond to a real world setting, where we might want to assign cell type and states to each sample or identify outliers. However, we value your concern and added an experiment according to your suggestion in the upcoming revision. SLLE-1 struggles similarly as ctSNE with this data, resulting in an embedding that does not reveal any information (a Gaussian blob).
>
> - Figure 3(a) is there to visualize how the vanilla embedding without prior looks like, and hence illustrate what the information is that we provide as prior to the methods. Methods 3(b) and (c) are important to compare, as they contain the embedding of the competing methods and show what novel information is revealed by the new embedding. In the upcoming draft we provide the standard UMAP embedding (3(a)) with coloring according to batch ID in the appendix.
>
> - For the coloring, we simply assigned colors from a color gradient to the ids specified by the authors of the corresponding single cell paper. The question whether close-by batch IDs are expected to be similar is biologically very interesting, and might be interesting for further practical analysis. As this is a proof of concept for a general method, we did not consider a more in-depth analysis of the data at hand.

---

> > ### Comment · AnonReviewer2 · 2020-11-22
> > **Thank you for the response**
> >
> > I have read the author response. Overall my evaluation currently still remains at the same level.
> >
> > "None of the available implementations for SLLE and ctSNE can deal with distance matrices as input." seems like a too brief dismissal of SLLE and ctSNE, at least the mathematics of ctSNE look to me like it can use distance matrices (unless I have missed something in the definition of that method), could a simple code change allow current implementations to be used?
> >
> > To me, the issue of bias in the flower experiment remains: it is not only that the color information was left in (I understand the motivation for that), but that the way the input distance is a linear sum of distances including color-distance: since this exactly matches the proposed distance-editing by subtracting the color-distance, it is not so surprising that such distance editing works for the data.

---

### Official Review · AnonReviewer3 · 2020-11-01
**Factoring out prior knowledge in dimensionality reduction**

**Rating:** 5
**Confidence:** 4

**Review:**

The paper presents JEDI and CONFETTI, two approaches to the task of factoring prior knowledge when creating low dimensional embeddings for data exploration. The authors allow the prior knowledge to be specified as distances based on an information theoretic extension to tSNE and UMAP.

The task is interesting and paper is well-presented. The theory section is a bit weak, although it provides an intuition of why those specific extra terms were added to the optimization problems.

I found the experiments interesting -- the three case studies are well thought out. However, I find the comparison with the competitors unfair in most cases, as the information that it is provided to those is quite different. In this case, I think the paper would benefit from not comparing to those methods but to use simple baselines of projections that de-bias the embeddings to illustrate the advantages of using either JEDI or CONFETTI.

Finally, I would like to have seen a discussion on how realistic it is that the prior knowledge would come in the form expected by the authors in real-world problems.

---

> ### Author Response · Authors · 2020-11-17
> **Answer to concerns of Reviewer #3**
>
> Dear Reviewer #3, first of all thank you for the constructive critique. The experimental setup is indeed inherently unfair. In the synthetic experiments, for example, we give our competitors a big advantage by providing the *ground truth* as prior knowledge, while our method has to make do with the distances obtained from the initial tSNE embedding. The problem is that no other method can remove prior information *beyond* labels, and to allow for meaningful comparison we therefore have to go out of our way, and re-cast tasks in such a way that existing methods have a chance at solving it (e.g. by providing labels obtained through clustering of the prior for single cell data). The experiments show that it is not possible to solve the core task satisfactorily with existing methods and simple transformations of the prior (e.g. through clustering).
>
> Regarding your comment on the discussion, yes that is an important point, that we could not sufficiently address in the original draft. In the upcoming revision we will add a section on this to the discussion. To answer your question directly, we think that prior knowledge in the form of distances or similarities is one of the most common and natural forms that a prior can take: examples include Euclidean distances derived from (single cell) gene expression data, distances between nodes in a social network, or even more general, kernel-based distances between strings, graphs, or other unstructured data – all of which can only be expressed to some extent with simple labels. In case of Facebook data in the ctSNE paper, for example, they use a labeling strategy based on geographic location (ZIP codes) as prior, which means that the relative distance is completely ignored. A more natural prior would be the (geodesic) distance between cities or areas (corresponding to the ZIP codes), which captures that interests of a person from New York compared to a person from New Jersey versus a person from Miami might be much closer. Sadly, their data was not available to us to allow this experiment.

---

### Decision · Program_Chairs · 2021-01-07
**Final Decision**

**Decision:**

Reject

**Comment:**

A method is proposed for removing prior knowledge, presented as a
distance matrix, from low-dimensional embeddings, to focus them on
what is new.

The task of visualizing novely in data is interesting and good
solutions would potentially be highly useful.

The proposed method essentially substracts a distance matrix from
another. While this is sensible, it is not completely clear in what
sense this is _the_ right solution for what the embeddings will be
used for.

In final discussions among the reviews, the main remaining concerns
were considered severe: comparisons to other methods being limited,
and possible problems in one of the experiments.